🔓 | **Open Peer Review** | Environmental Microbiology | Research Article

# Transcriptomic response of the picoalga *Pelagomonas calceolata* to nitrogen availability: new insights into cyanate lyase function

Nina Guérin,[1,2] Chloé Seyman,[1,2] Céline Orvain,[1,2] Laurie Bertrand,[1,2] Priscillia Gourvil,[3] Ian Probert,[3] Benoit Vacherie,[4] Élodie Brun,[4] Ghislaine Magdelenat,[4] Karine Labadie,[4] Patrick Wincker,[1,2] Adrien Thurotte,[1,2] Quentin Carradec[1,2]

**ABSTRACT** Cyanate ($OCN^-$) is an organic nitrogen compound found in aquatic environments potentially involved in phytoplankton growth. Given the prevalence and activity of cyanate lyase genes in eukaryotic microalgae, cyanate has been suggested as an alternative source of nitrogen in the environment. However, the conditions under which cyanate lyase is expressed and the actual capacity of microalgae to assimilate cyanate remain largely underexplored. Here, we studied the nitrogen metabolism in the cosmopolitan open-ocean picoalga *Pelagomonas calceolata* (Pelagophyceae and Stramenopiles) in environmental metatranscriptomes and transcriptomes from culture experiments under different nitrogen sources and concentrations. We observed that cyanate lyase is upregulated in nitrate-poor oceanic regions, suggesting that cyanate is an important molecule contributing to the persistence of *P. calceolata* in oligotrophic environments. Non-axenic cultures of *P. calceolata* were capable of growing on various nitrogen sources, including nitrate, urea, and cyanate, but not ammonium. RNA sequencing of these cultures revealed that cyanate lyase was downregulated in the presence of cyanate, indicating that this gene is not involved in the catabolism of extracellular cyanate to ammonia. Based on environmental data sets and laboratory experiments, we propose that cyanate lyase is important in nitrate-poor environments to generate ammonia from cyanate produced by endogenous nitrogenous compound recycling rather than being used to metabolize imported extracellular cyanate as an alternative nitrogen source.

**IMPORTANCE** Vast oceanic regions are nutrient-poor, yet several microalgae thrive in these environments. While various acclimation strategies to these conditions have been discovered in a limited number of model microalgae, many important lineages remain understudied. Investigating nitrogen metabolism across different microalga lineages is crucial for understanding ecosystem functioning in low-nitrate areas, especially in the context of global ocean warming. This study describes the nitrogen metabolism of *Pelagomonas calceolata*, an abundant ochrophyte in temperate and tropical oceans. By utilizing both global scale *in situ* metatranscriptomes and laboratory-based transcriptomics, we uncover how *P. calceolata* adapts to low-nitrate conditions. Our findings reveal that *P. calceolata* can metabolize various nitrogenous compounds and relies on cyanate lyase to recycle endogenous nitrogen in low-nitrate conditions. This result paves the way for future investigations into the significance of cyanate metabolism within oceanic trophic webs.

**KEYWORDS** nitrogen metabolism, microalgae, *Pelagomonas*, transcriptomics, cyanate lyase

**Peer Reviewer** Nastasia Johana Freyria, McGill University, Montréal, Québec, Canada

Address correspondence to Quentin Carradec, qcarrade@genoscope.cns.fr.

The authors declare no conflict of interest.

See the funding table on p. 18.

N itrogen is essential to many biological processes, such as photosynthesis, amino acid, and nucleic acid biosynthesis; thus, its low bioavailability in the oceans

impacts the growth of primary producers (1). The primary sources of fixed nitrogen for phytoplankton are inorganic ammonium ($NH_4^+$), nitrite ($NO_2^-$), and nitrate ($NO_3^-$) (2). Nitrate is the most abundant nitrogenous compound, whereas nitrite and ammonium are typically less abundant. Nevertheless, ammonium is considered a preferred source of nitrogen for phytoplankton due to its lower energy cost during assimilation (3). The majority of oceanic surface waters are depleted in inorganic nitrogen compounds because of uptake by photosynthetic organisms in the photic zone (2). One consequence of ocean warming is enhanced stratification, which reduces the supply of nutrients to the euphotic zone (4). By the end of the 21st century, an average of $1.06 \pm 0.45$ mmol $m^{-3}$ decrease in nitrate concentration in surface waters is projected under the IPCC high-emission scenario SSP5-8.5 (5).

Due to the variability in inorganic nitrogen compound concentrations, photosynthetic organisms have evolved a variety of strategies to cope with the temporal and spatial fluctuations in the availability of nitrate, nitrite, and ammonium. Optimization of inorganic nitrogen uptake can be achieved by regulating the expression of transporters in several phytoplankton groups, notably prasinophytes and diatoms (6). Storage and recycling of nitrogen-rich proteins are important strategies employed by diatoms (7, 8). Various phytoplankton can metabolize dissolved organic nitrogen (DON) compounds, such as urea, purines, or amino acids (3). In the ocean, the concentration of DON, often higher than that of dissolved inorganic nitrogen (DIN), contributes to the autotrophic production and growth of primary producers, especially in coastal and estuarine environments (9). DON compounds have been shown to support the growth of microalgae in low N-conditions. For example, urea can be used as a nitrogen source by many phytoplankton groups, such as diatoms, dinoflagellates, and the bloom-forming pelagophyte *Aureococcus anophagefferens* (10–12). Diatoms such as *Phaeodactylum tricornutum* and *Thalassiosira pseudonana* possess urea transporters that are upregulated in N-limited conditions (8, 13). The coccolithophore *Emiliania huxleyi* can use a broad range of organic nitrogen sources, including urea, hydroxyurea, hydroxanthine, purines, and small amides such as acetamides and formamides (14).

The cyanate ion ($OCN^-$), which is the smallest nitrogenous organic compound, was originally described as a toxic molecule altering the structural and functional properties of proteins through carbamylation (15). In the oceans, cyanate originates from terrestrial inputs, spontaneous decomposition of carbamoyl-phosphate (CP) or urea released by zooplankton or senescent phytoplankton, as well as via the photochemical degradation of dissolved organic matter (16, 17). Cyanate concentrations up to 45 nM have been recorded in subsurface ocean waters (18). Growth on cyanate as the sole nitrogen source was first described in *Escherichia coli*, then in cyanobacteria, the ammonia-oxidizing archaea *Nitrososphaera gargensis*, several yeasts, and the mixotrophic dinoflagellate *Prorocentrum* (19–24). In bacteria, the cyanate lyase enzyme (cynS gene) catalyses the bicarbonate-dependent breakdown of cyanate to ammonia and carbon dioxide (25). Cyanate lyase homologs are present in the genome of many organisms, including dominant eukaryotic phytoplankton lineages (26). In the environment, the expression of cyanate lyase in most prokaryotic and eukaryotic phytoplankton is increased in N-limited environments, suggesting that these organisms might use cyanate as an alternative nitrogen source (26–29). The $^{15}N$ stable isotope probing revealed that cyanate uptake could account for up to 10% of total nitrogen uptake in natural communities from the offshore oligotrophic Atlantic, particularly in surface waters (17). Since cyanate lyase expression is as important as urease expression in the environment, it has been suggested that cyanate has a crucial ecological role (26). However, the uptake of cyanate in eukaryotic phytoplankton has only been demonstrated in *Prorocentrum*. A recent study reported that cyanate enrichment in natural phytoplankton populations induces the growth of the picocyanobacterium *Synechococcus*, but not that of eukaryotic phytoplankton (30). This leads us to question the role of cyanate lyase in organic nitrogen assimilation in photosynthetic eukaryotes.

In order to disentangle the endogenous role of cyanate lyase in microeukaryotes and more generally to develop a better understanding of eukaryotic phytoplankton acclimation to varying nitrogen availability in the environment, we used *Pelagomonas calceolata*, a cosmopolitan Pelagophyceae, as a model for pelagic phytoplankton (31, 32). Pelagophyceae are a diverse group of marine microalgae comprising 4 families and 23 genera (33). They cover all oceanic basins, from polar waters to tropical oceans (34). Most species have been described in coastal environments, with some forming brown tides (35). The ability to consume organic nitrogen has been shown to contribute to *Aureococcus* blooms (36). Among the few pelagophytes present in the open ocean, *Pelagomonas* is the dominant taxa and is widely distributed in temperate and tropical oceans (32). Environmental studies have shown that *Pelagomonas* present strong acclimation abilities, especially to iron and nitrate depletion (37, 38). The *P. calceolata* (strain RCC100) genome contains a large set of genes involved in nitrogen metabolism (32). The presence of genes coding for arginase, urease, and cyanate lyase may suggest the capacity for metabolism of organic nitrogen compounds. In the environment, *P. calceolata* upregulates nitrogen ion transporters, nitrate and nitrite reductases, glutamine synthetases (GSs), nitrate/nitrite sensing proteins, and cyanate lyase in low-N environments (32, 38).

Here, we cultivated two *P. calceolata* strains (RCC100 and RCC697) under reduced nitrate concentration and tested the effect of several inorganic (nitrate and ammonia) and organic (urea and cyanate) sources of nitrogen on *P. calceolata* growth. To evaluate *P. calceolata's* response to different growth conditions, we conducted RNA sequencing and identified differentially expressed genes (DEGs). We then integrated our findings with a differential analysis of *P. calceolata* gene expression levels from environmental metatranscriptomes collected during the *Tara* Oceans expedition and correlated these results with *in situ* nitrate concentrations.

## MATERIALS AND METHODS

### Environmental metatranscriptomes and associated metadata

Metatranscriptomic data sets from the *Tara* Oceans and *Tara* Polar Circle expeditions (39) were used to detect the *in situ* gene expression of *P. calceolata*. All available data sets from seawater samples in the photic zone (73 surface and 51 deep-chlorophyll maximum samples) and from two size fractions (80 from the 0.8–5 µm fraction and 44 from the 0.8–2,000 µm fraction) were selected. Metatranscriptomic reads were aligned on the 16,667 predicted mRNA sequences of the *P. calceolata* genome (ENA, PRJEB47931) with bwa-mem2 version 2.2.1 with default parameters (40). Reads that aligned to the *P. calceolata* genome with >95% identity over 80% of read length were selected. Nuclear genes covered by at least 10 reads in a minimum of 10 samples were retained. To eliminate putative cross-mapped genes (i.e., highly conserved genes that probably aggregate reads from other organisms), genes detected in >90% of samples (including those where *P. calceolata* is not present) were removed. Finally, samples with >75% of *P. calceolata* genes were kept for the next steps. The environmental parameters measured during the expedition are available in the Pangaea database (https://www.pangaea.de/) (41). Nitrate concentrations were calculated from *in situ* sensor (SATLANTIC) data, calibrated using water samples. Samples were considered "low-nitrate" if they contained <2 µM of nitrate. Differential expression analyses were conducted with the DESeq2 package version 1.32.0 under R version 4.1.1 across the 15,617 genes and the 112 environmental samples with available *in situ* nitrate concentration measurements (42). Pairwise comparisons were performed across 43 "high-nitrate" samples (nitrate concentration >2 µM) and 69 "low-nitrate" samples (nitrate concentration <2 µM), and with the function DESeq with default parameters, log2 fold change (log2FC) values were calculated with the lfcShrink function. Genes with a *P* value < 0.01 and a log2FC >2 or <−2 were considered as differentially expressed.

## P. calceolata cultures in different nitrogen conditions

P. calceolata strains RCC100 and RCC697 (obtained from the Roscoff Culture Collection: www.roscoff-culture-collection.org) were cultivated in non-axenic conditions in artificial seawater (ASW) supplemented by L1 medium (as described in Reference 43). ASW was prepared by dissolution of 24.55 g of sodium chloride (NaCl), 0.75 g of potassium chloride (KCl), 4.07 g of magnesium chloride hexahydrate (MgCl$_2$·6H$_2$O), 1.11 g of calcium chloride (CaCl$_2$), 2.95 g of magnesium sulfate (MgSO$_4$), and 0.21 g of sodium bicarbonate (NaHCO$_3$) in 1 L of sterile distilled water. A volume of 1 mL of trace metals, vitamins, and nutrients from the Bigelow L1 Medium Kit (MKL150L) was added to attain the following concentrations: 882 µM sodium nitrate (NaNO$_3$), 36.2 µM monosodium phosphate (NaH$_2$PO$_4^-$), and 106 µM sodium silicate (Na$_2$SiO$_3$). Cultures were maintained at 20°C under a 12:12 h light-dark photoperiod and a blue light (while LEDs were covered by a blue filter: LEE FILTERS, 183 Moonlight blue) at an intensity of 20 µmol m$^{-2}$ s$^{-1}$ of photosynthetic photons. The non-flagellated P. calceolata strain (RCC697) was maintained on an orbital shaker (Kühner) at 150 r min$^{-1}$. For the nitrate depletion experiment, the nitrate concentration in the L1 medium was reduced to 441, 220, 110, or 50 µM. For the cyanate and the ammonium experiments, nitrate was replaced by potassium cyanate (KOCN) or ammonium chloride (NH$_4$Cl) at the same concentration (882 µM). For the urea experiment, nitrate was replaced by urea (CH$_4$N$_2$O) at 441 µM to ensure that the nitrogen atom concentration remained consistent across all conditions. Prior to the growth experiment under cyanate, the P. calceolata RCC100 culture was made axenic with a mix of antibiotics (Spectinomycin [50 µg/mL], Neomycin [100 µg/ml], and Carbenicillin [30 µg/mL]). Axenicity was verified on marine broth plates (Difco 2216). No reduction in the growth of axenic versus non-axenic cultures was observed under standard culture conditions. During this experiment, P. calceolata cells were counted daily with a flow cytometer (Cytoflex, Beckman Coulter Life Sciences). Cell counts were fitted to a logistic growth curve with the R package Growthcurver (version 0.3.1). The r statistics given by GrowthCurver correspond to the growth rate that would occur if there were no restrictions imposed on total population size.

RCC100 and RCC967 were grown in each condition tested during an acclimation phase lasting a minimum of 8 days. Cultures were then diluted in fresh medium in triplicate and grown for 5–9 days until they reached a minimum concentration of 10 million cells per mL. Growth and fluorescence were monitored daily using a Qubit Instrument (Invitrogen Qubit 3 Fluorometer Q33216, blue excitation at 470 nm, far red emission 665–720 nm), and the cells were counted on a Thoma cell (Marienfeld Thoma counting chamber, depth 0.1 mm, 0640710) under a microscope on the day of harvesting. P. calceolata cells were harvested by filtration through 1.2 µm mixed cellulose-ester membrane filters (MF-Millipore, rawp04700) with a peristaltic pump (SFP-100), then transferred into 15 mL tubes, flash-frozen in liquid nitrogen, and stored at −80°C until RNA extraction.

## Bacteria isolation and identification

To isolate bacteria present in P. calceolata cultures, solid marine broth medium (MB 2216, NutriSelect) containing 15 g/L of agar (BD Difco Bacto Dehydrated Agar) was prepared on Petri dishes. A volume of 100 µL of a non-axenic culture of P. calceolata (strain RCC100) in the exponential growth phase was incubated at 37°C in the dark. After 3 days, 15 brown and 24 white colonies were picked and transferred in 20 µL of sterile water. The bacterial 16S rRNA were amplified using 12.5 µL of 2× KAPA Library Amplification ReadymIX (Roche) and 1 µL (10 µM) of universal forward 27F (AGAGTTTGATCMTGGCTCAG) and reverse 1492R (TACGGYTACCTTGTTACGACTT) primer sequences (44). The following polymerase chain reaction (PCR) conditions were used: initial denaturation (95°C 5 min), 30 cycles (95°C 5 min; 55°C 20 s; and 72°C 1 min), and final extension 72°C 5 min. Amplification products were purified with 1.8 volumes of Ampure Xp beads (Beckman Coulter A63880) and then sequenced with the same primers

on an ABI3730 sequencer device (Applied Biosystems). The closest bacterial species to each 16S sequence was identified by nucleotide BLAST against NR (on the NCBI website).

## RNA extraction and sequencing

Flash-frozen filters were vortexed in QIAzol, and RNA was then extracted using RNeasy Plus Universal Mini Kits (Qiagen, Ref 73404) following the manufacturer's instructions. All extracted RNA samples were treated with 6U of TURBO DNase (2 U/µL) (Thermo Fisher Scientific, Ref. AM2238), then purified with RNA Clean and Concentrator-5 kit (Zymo Research, Ref. ZR1016), keeping only large RNA fractions (>200 nt) for RNAseq library preparation. A total of 100 ng of treated RNA was used to produce Illumina libraries (Illumina Stranded mRNA Prep, Ligation). Briefly, poly(A)+ RNAs were selected with oligo(dT) beads, chemically fragmented by divalent cations under high temperature, converted into single-stranded cDNA using random hexamer priming, followed by second-strand synthesis and 3′-adenylation. A pre-index anchor was ligated, and a PCR amplification step with 15 cycles was conducted to add 10 bp unique dual index adapter sequences (IDT for Illumina RNA UD Indexes, Ligation). All libraries were quantified using Qubit dsDNA HS Assay measurement. A size profile analysis was performed in an Agilent 2100 Bioanalyzer (Agilent Technologies, Santa Clara, CA, USA). The library preparation failed for one sample of RCC697 (200 µM NO$_3$). Libraries were sequenced in 2 × 150 bp on an Illumina NovaSeq 6000 sequencer (Illumina, San Diego, CA, USA) in order to obtain 50 million paired-end reads. After Illumina sequencing, an in-house quality control process was applied to the reads that passed the Illumina quality filters (39). Briefly, Illumina sequencing adaptors and primer sequences were removed, then low-quality nucleotides ($Q < 20$) were discarded from both ends of the reads. Sequences between the second unknown nucleotide (N) and the end of the read were also trimmed. Reads shorter than 30 nucleotides were discarded after trimming with an adaptation of the fastx_clean tool (https://www.genoscope.cns.fr/fastxtend/). In the last step, reads that were mapped to the Enterobacteria phage PhiX174 genome (GenBank: NC_001422.1) were discarded using bowtie2 v2.2.9 (-L 31-mp 4-rdg 6,6-local-no-unal) (45). Remaining rRNA reads were removed using SortMeRNA v2.1 and SILVA databases version 119 (46, 47).

## Analysis of gene expression levels in different nitrogen conditions

As for the analysis of environmental metatranscriptomes, RNAseq reads were aligned with bwa-mem2 version 2.2.1 on the *P. calceolata* genome strain RCC100, the only genome available for this species. Reads with a minimal size of 50 bp and aligned with >95% identity over 80% of the read length were selected. We detected 16,659 out of 16,667 genes in at least one sample. Only nuclear genes were retained, with gene expression levels normalized in transcripts per kilobase per million mapped reads (TPM). Pearson's correlation matrices were computed between triplicates and across conditions, based on the gene expression normalized by TPM (cor function of R package stats version 4.1.1). Hierarchical clustering of Euclidean distance between samples was performed with the dist and hclust functions of the stats package on R version 4.1.1. Differential expression analysis on the transcriptomic samples was carried out in the same way as for the metatranscriptomic DESeq2 analysis above. Identification of DEGs between control and test samples was performed by pairwise comparisons across the standard condition (882 µM NO3) and low-nitrate conditions (220 µM or 441 µM NO3) on the one hand, and across the standard condition and changing nitrogen sources (882 µM ammonium, 882 µM cyanate, and 441 µM urea) on the other hand. Genes presenting a *P* value < 0.01 and a log2FC >2 or <−2 were considered as differentially expressed. We specifically looked for log2FC of genes identified as involved in the nitrogen cycle in our previous study (32). Figure 1 to 5 were generated with ggplot2 version 3.5.0 except for the Euler diagrams that were made with eulerr version 7.0.2 and graphics version 4.1.1.

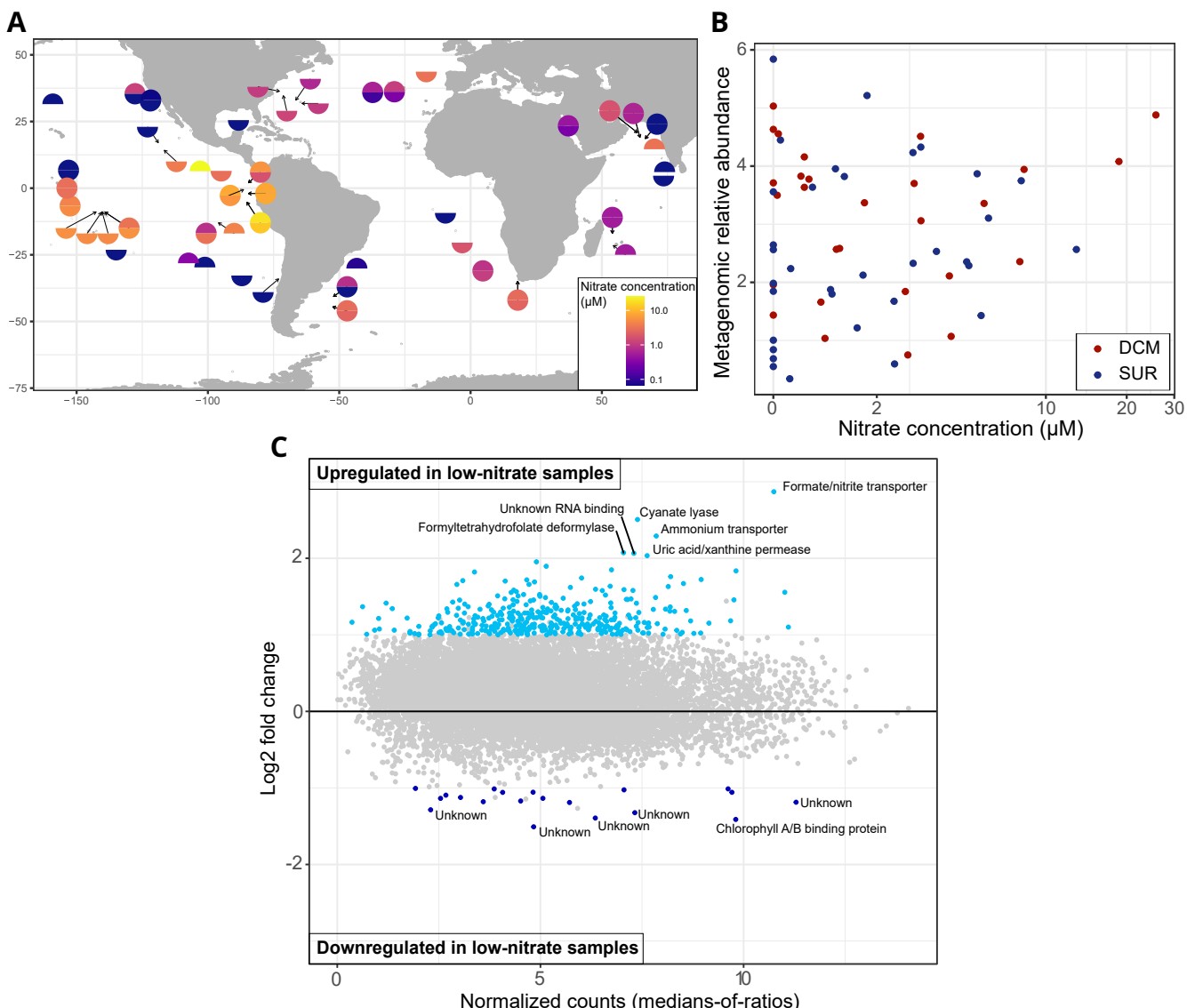

**FIG 1** Abundance and transcriptomic response of *P. calceolata* to environmental nitrate concentrations. (A) Nitrate concentrations were measured during the *Tara* Oceans expedition. The color code indicates nitrate concentrations in μmol/L for surface (SUR) and deep chlorophyll maximum (DCM) samples in the upper and lower parts of each dot, respectively. (B) Relative abundance of *P. calceolata* in *Tara* samples estimated from metagenomics reads according to the concentration of nitrate (μM). (C) *P. calceolata* gene expression levels between low-nitrate ($NO_3^- < 2$ μM, $n = 69$) and high-nitrate samples ($NO_3^- > 2$ μM, $n = 43$). Log2FC between low-nitrate and high-nitrate samples are given according to their mean expression level (normalized with DESeq2). Differentially expressed genes with $P$ value < 0.01 and log2FC > 1 or <−1 are colored in blue.

## Functional re-annotation of *P. calceolata* genes

Functional annotation of *P. calceolata* genes was updated for this study using Inter-ProScan v5.61.93.0 (48) and the following protein domain databases: Pfam, Gene3D, TIGRfam, SMART, CDD, and gene ontology. All matches with a $P$ value below $1 \times 10^{-5}$ were retained. A protein alignment against the NR database (24-08-2023 version) was performed with diamond v2.1.2 (49). The best match was retained if the $e$-value was below $1 \times 10^{-5}$. The HMM search tool KofamKoala v1.3.0 was used to identify KEGG

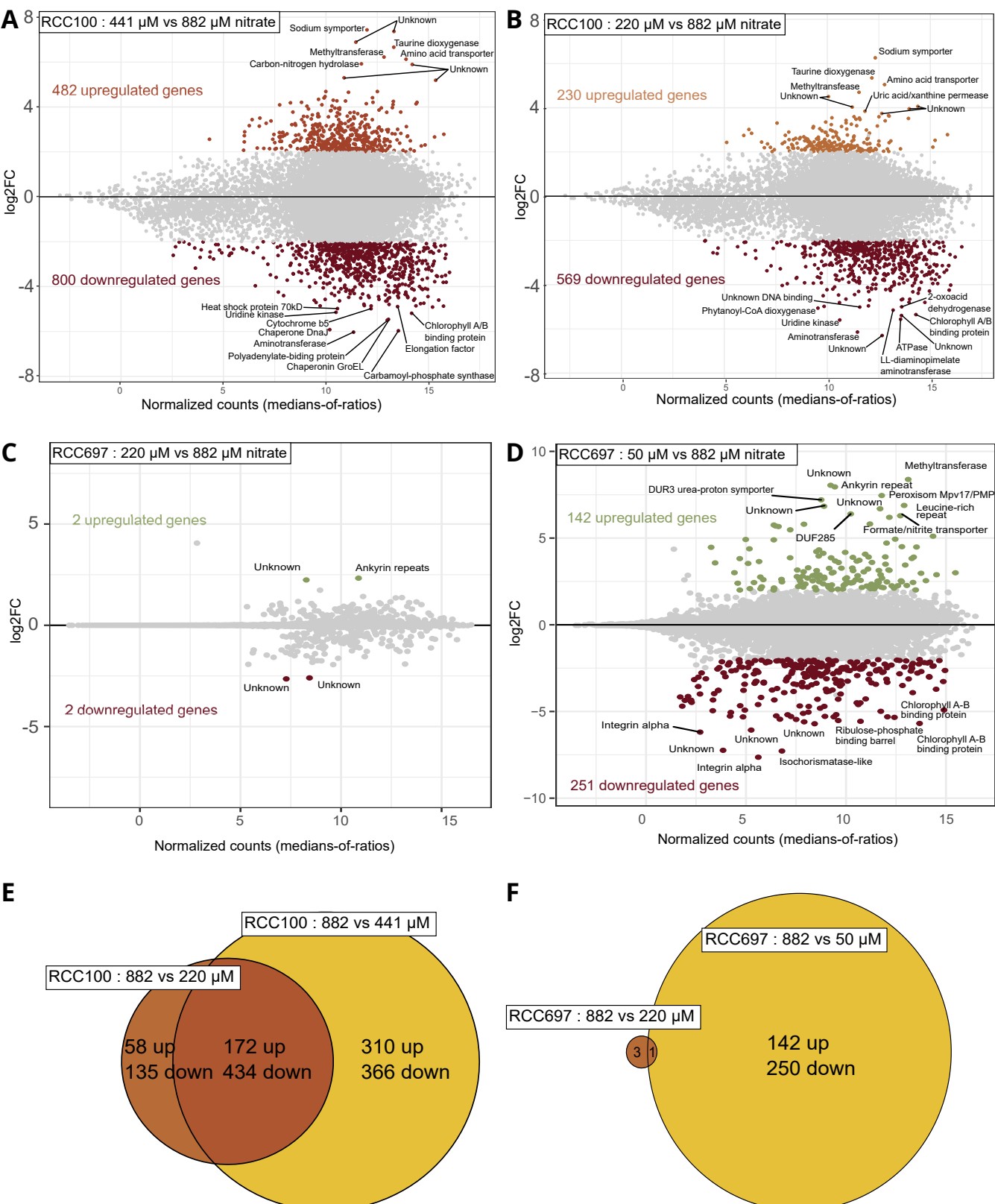

**FIG 2** Transcriptomic response of *P. calceolata* to low-nitrate culture conditions. (A–D) Differentially expressed genes of *P. calceolata* RCC100 in 441 µM (A) and 220 µM (B) nitrate and RCC697 in 220 µM (C) and 50 µM (D) compared to 880 µM nitrate. Genes with *P* value < 0.01 and log2FC > 2 are colored. (D and E) Euler diagram of DEGs in RCC100 (E) and RCC697 (F). The number of upregulated and downregulated genes is indicated. DEG, differentially expressed gene.

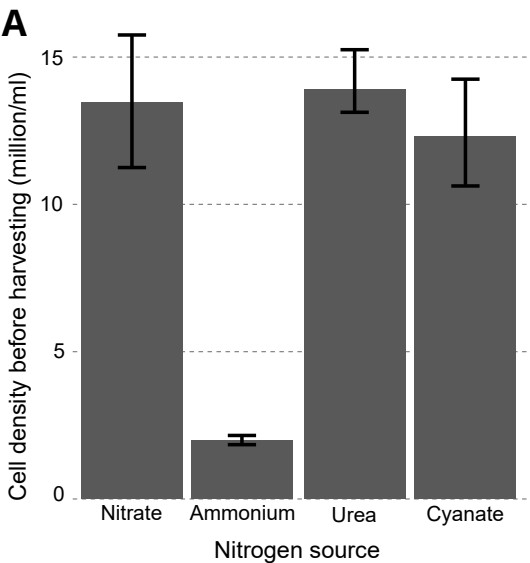
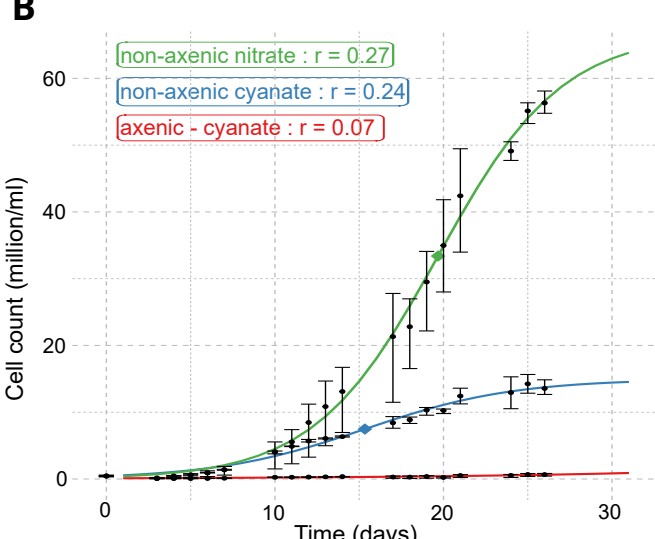

**FIG 3** *P. calceolata* growth with different nitrogen sources. (A) Average *P. calceolata* density (cell/mL) before harvesting for RNA extraction (at the beginning of the exponential growth phase). Cell counts were measured on a Thoma counting chamber under light microscopy. (B) *P. calceolata* growth curves were cultivated with nitrate, cyanate, or cyanate after antibiotic treatments to remove the bacterial community. Cell counts were estimated with a flow cytometer. Experiments in (A and B) were carried out in triplicate, and error bars indicate minimum and maximum values for each condition.

Orthologues (version of November 2023) (50). Annotations with an *e*-value $<1 \times 10^{-5}$ and a score above the HMM threshold were retained. Homologies with protein clusters of the Eggnog database were identified with the eggnog-mapper tool version 2.1.12 using the *very-sensitive* mode and diamond aligner (51, 52). For the prediction of protein localization, DeepLoc version 2.0 (53) and TargetP version 2 in eukaryote mode (54) were used. This methodology was applied on the 16,667 gene models, translated in the six frames in full with the transeq function of Emboss version 6.6 and on the protein predicted by Gmove (55). If functional annotations were identified on several frames, the frame with the best score was retained. This new version of the functional annotation of *P. calceolata* genes is available on GitHub https://github.com/institut-de-genomique/PelagomonasNitrogenMetabolism/.

## RESULTS

### *In situ* gene expression levels of *P. calceolata* according to nitrate concentration

To determine the *in situ* response of *P. calceolata* to low nitrogen concentration, we used all metatranscriptomes rom the *Tara* Oceans data sets (39). We aligned metatranscriptomic reads on the predicted mRNAs of the *P. calceolata* RCC100 genome and selected 124 *Tara* samples where at least 75% of the genes were detected (at least one read aligned with more than 95% identity over 80% of its length). Of the 112 samples with available *in situ* nitrate measurements, 69 have nitrate concentrations below 2 µM and are considered as "low-nitrate", while 43 have nitrate concentrations above 2 µM and are considered as "high-nitrate" (Fig. 1A; Table S1). We observed no significant correlation between nitrate concentrations and the relative abundance of *P. calceolata* (Fig. 1B; Pearson $r = 0.2$, *P* value = 0.11).

Differential expression analyses between "high-nitrate" and "low-nitrate" environments revealed 375 genes significantly upregulated in low-nitrate samples (*P* value < 0.01), 6 of which had a log2FC greater than 2 (Fig. 1C; Table S2). Two of these genes are involved in inorganic nitrogen transport across cell membranes, a formate or nitrite transporter (PF01226 domain) and an ammonium transporter (PF00909 domain) as previously described (32). In addition, a purine transporter (uric acid/xanthine permease,

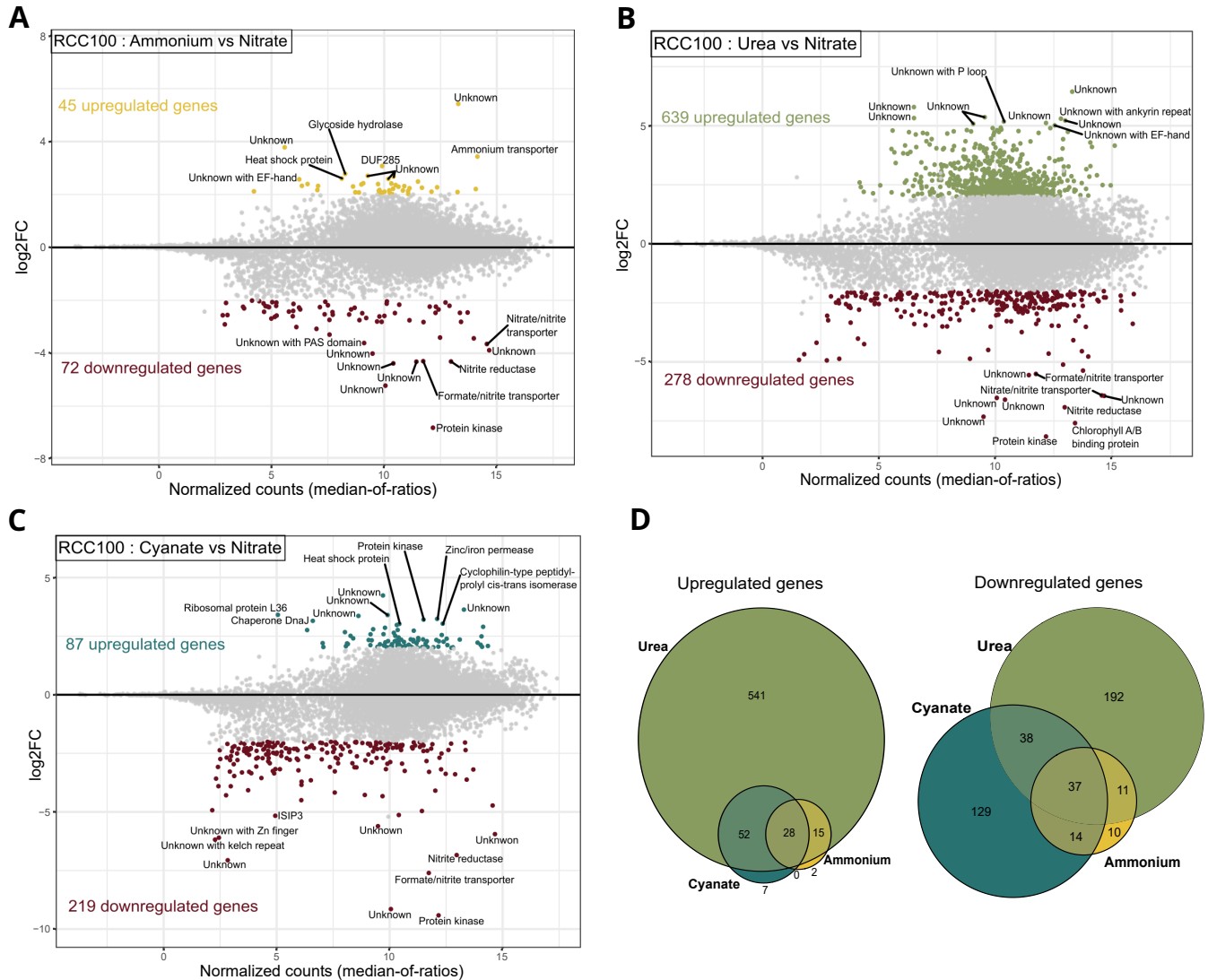

**FIG 4** Transcriptomic response of *P. calceolata* RCC100 cultivated with different nitrogen compounds. (A–C) Differentially expressed genes in 882 µM ammonium (A), 441 µM urea (B), and 882 µM cyanate (C) compared to 882 µM nitrate. Genes with *P* value < 0.01 and log2FC > 2 are colored. The functions of the top 10 genes upregulated or downregulated are indicated. (D) Euler diagrams of genes upregulated (left) or downregulated (right) in at least one of the alternative nitrogen sources.

K23887 domain), a formyltetrahydrofolate deformylase (10-FDF; K01433-EC 3.5.1.10), an enzyme acting on carbon-nitrogen bonds and potentially related to ammonium recycling from glycine, as well as an unknown gene carrying an RNA-binding domain of the Pumilio family (K17943 domain) were upregulated in low-nitrate environments. The cyanate lyase (PF02560 domain) was the second most upregulated gene in the *P. calceolata* genome in low-nitrate environments. We also noted the slight upregulation of a nitrate/nitrite transporter (K02575 domain), a carbon nitrogen hydrolase (PF00795 domain), and a dipeptidase (K08659 domain), suggesting active recycling of nitrogen-rich molecules. Only 20 genes were slightly downregulated in low-nitrate environments with a log2FC between 1 and 1.5, most of which (14 genes) have unknown functions. Three Light Harvesting Complex proteins (PF00504 domain) were downregulated in low-nitrate environments, suggesting reduced chloroplast activity (Table S2).

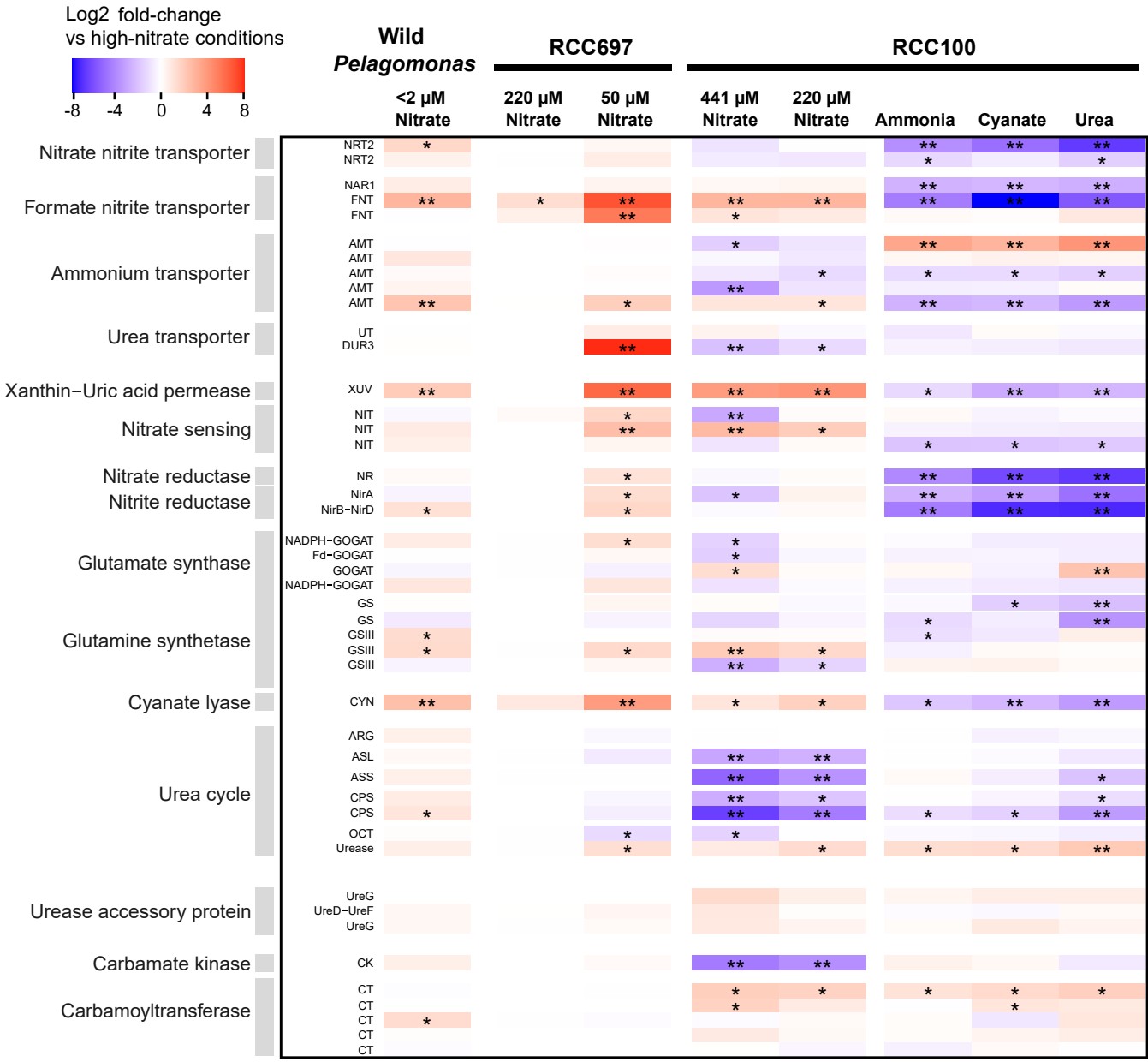

**FIG 5** Differential expression of *P. calceolata* genes involved in nitrogen metabolism. The column named wild *Pelagomonas* represents DEGs in the *Tara* Oceans samples between low-nitrate (<2 µM) and high-nitrate (>2 µM) environments. *log2 FC >1 or <−1 and *P* value < 0.01. **log2 FC >2 or <−2 and *P* value < 0.01. DEG, differentially expressed gene.

## Gene expression variations under low-N conditions

To estimate acclimation capacities of *P. calceolata* to low-nitrate conditions, we cultivated two strains of this species complex. RCC100 (=CCMP1214), isolated in the Pacific Ocean in 1973, is flagellated with a cell size of 1.5–2 µm, while RCC697, isolated in the Indian Ocean in 2003, is significantly larger with a cell diameter of 3 µm and is non-flagellated (Fig. S1A and B). Since their isolation, both strains have been maintained in the Roscoff Culture Collection at 20°C. These strains were cultivated with different nitrate concentrations ranging from 50 to 882 µM for a minimum of 3 weeks with weekly dilutions in fresh media (Fig. S1C and D). The minimal nitrate concentration to observe cell growth and reach a sufficient number of cells for RNA extraction was 220 µM for RCC100 and 50 µM for RCC697.

*P. calceolata* cells were harvested in the exponential growth phase for RNA extraction in the following conditions: 220, 441, and 882 µM for RCC100 and 50, 220, and 880 µM for RCC697. Between 29.9 and 84.5 million paired-end RNA reads were sequenced for each sample, then aligned to the predicted mRNAs of the *P. calceolata* RCC100 genome. The two strains are genetically close with an average of 96.0% of nucleotide identity for 86% of RCC100 genes covered by at least one RCC697 read (Table S3). Correlations of gene expression levels within triplicates were higher (Pearson's *r* coefficients > 0.95) than correlations between conditions (Pearson < 0.81), with the exception of RCC697 882 µM versus RCC697 220 µM nitrate, which were highly correlated (Pearson > 0.92; Fig. S2). These correlations indicate that all tested conditions induced a transcriptomic response of *P. calceolata*, except the reduction to 220 µM nitrate in RCC697.

Differential expression analyses were computed between the high-nitrate condition (882 µM) and each tested condition for both strains. For RCC100, 1,282 genes were differentially expressed with 441 µM nitrate and 799 were differentially expressed with 220 µM nitrate (Fig. 2A and B). A large proportion of these DEGs were common between the two reduced nitrate conditions (604 DEGs, 172 upregulated and 434 downregulated), indicating that the cells were already acclimated to low-nitrate in the 441 µM experiment (Fig. 2E). For RCC697, only four genes were differentially expressed in the 220 µM condition, indicating that this nitrate reduction did not strongly affect this strain in contrast to RCC100. With only 50 µM nitrate, RCC697 exhibited an important transcriptomic response with 393 DEGs (251 downregulated and 142 upregulated; Fig. 2C, D and F). RCC100 and RCC697 had a similar response to the limitation of nitrate, with a total of 97 common DEGs (Table S4).

The six upregulated genes in low-nitrate environments were also upregulated in our culture experiments when nitrate concentration was limited (Table 1). We note that the ammonium transporter was slightly differentially expressed only in the strongest nitrate depletion conditions (RCC697 50 µM). Other DEGs involved in nitrogen metabolism are described in the following paragraphs.

## Gene expression levels of *P. calceolata* RCC100 cultivated with different nitrogen compounds

*P. calceolata* RCC100 was cultivated with nitrate (882 µM, standard conditions), ammonium (882 µM), urea (441 µM), or cyanate (882 µM) for a minimum of 3 weeks with weekly transfer to fresh media. *P. calceolata* was able to grow with cyanate or urea as the sole source of nitrogen, but the growth was strongly limited with only ammonium as a nitrogen source (Fig. 3A). To test whether the growth under cyanate could involve consumption of cyanate by the bacterial community and then metabolite exchanges, we monitored RCC100 under cyanate in axenic versus non-axenic conditions (Fig. 3B). Growth under cyanate in axenic conditions was strongly reduced, indicating that RCC100 cannot metabolize cyanate without the bacterial community. To identify these bacteria, we isolated bacterial colonies on solid media from a non-axenic *P. calceolata* culture and sequenced the 16S rRNA of 39 colonies (see Materials and Methods). We identified 15

**TABLE 1** Differentially expressed genes in low-nitrate conditions in the environment and in culture[a]

| Gene | Function | Wild *Pelagomonas* | | *Pelagomonas* culture RCC100 | | | | *Pelagomonas* culture RCC697 | | | |
|------|----------|--------|--------|--------|--------|--------|--------|--------|--------|--------|--------|
| | | <2 µM | | 220 µM | | 440 µM | | 50 µM | | 220 µM | |
| | | L2FC | *P* value | L2FC | *P* value | L2FC | *P* value | L2FC | *P* value | L2FC | *P* value |
| Pca_2p03910 | Uric acid-xanthine permease | 2.03[b] | 5E−12 | 4.14[b] | 3E−48 | 3.86[b] | 2E−42 | 5.70[b] | 2E−36 | 0.01 | 1E+0 |
| Pca_4p09810 | Formate/nitrite transporter | 2.87[b] | 1E−34 | 2.87[b] | 3E−24 | 2.85[b] | 2E−24 | 6.29[b] | 7E−83 | 1.35 | 4E−3 |
| Pca_3p16260 | Cyanate lyase | 2.51[b] | 4E−19 | 1.82[b] | 8E−6 | 1.03 | 8E−3 | 3.82[b] | 8E−48 | 0.94 | 1E−2 |
| Pca_1p08820 | Ammonium transporter | 2.29[b] | 3E−11 | 1.06 | 1E−2 | 0.99 | 1E−2 | 1.91[b] | 2E−20 | 0.02 | 1E+0 |
| Pca_4p09790 | Formyltetrahydrate deformylase | 2.07[b] | 3E−13 | 2.56[b] | 5E−21 | 2.73[b] | 8E−24 | 4.50[b] | 1E−52 | 0.06 | 3E−1 |
| Pca_3p08530 | Pumilio RNA-binding repeat | 2.07[b] | 4E−14 | 2.19[b] | 1E−7 | 0.62 | 1E−1 | 2.54[b] | 1E−52 | 1.02[b] | 2E−6 |

[a]Log2 fold changes for each condition and each gene upregulated in low-nitrate environments are indicated. The complete list of DEGs in culture experiments is in Table S4.
[b]*P* values < 0.001.

colonies with identical 16S sequences sharing 99.6% of identity with two *Paracoccus* sp. strains (accessions KM083572.1 and NR_157753.1). The other 24 colonies showed 100% identity with *Marinobacter* sp. (accession MF401328.1). Several genomes available for these strains carry the cyanate lyase (CynS) and the cyanate transporter (CynX), strengthening the possibility that the phycosphere of *P. calceolata* converts the cyanate into a metabolite that is then consumed by the algae.

To determine the genes involved in the metabolic rewiring of non-axenic *P. calceolata* cultivated under different nitrogen compounds, we extracted and sequenced polyA+ RNAs in triplicate in the four conditions of Fig. 3A. We obtained between 45 million and 84 million paired-end reads for each sample that were aligned on the predicted mRNAs of the *P. calceolata* RCC100 genome (Table S3). For all conditions, triplicates were highly correlated, with Pearson correlations above 0.98 (Fig. S2B). Gene expression levels of nitrate, cyanate, and ammonium conditions were more similar (Pearson > 0.95) compared to urea (Pearson between 0.73 and 0.82).

In the ammonium condition, despite strongly impacted growth, only 72 genes were downregulated and 45 upregulated compared to the nitrate condition (Fig. 4A). In contrast, *P. calceolata* RCC100 exhibited good acclimation to cyanate and urea, and many genes were differentially expressed (917 and 306 DEGs, respectively; Fig. 4B and C; Table S5). Taken together, cyanate, urea, and ammonium conditions had 65 DEGs in common, 28 of which were upregulated and 37 downregulated (Fig. 4D; Table S6). Among the 37 downregulated genes, 8 are involved in nitrate transport and assimilation, indicating a global downregulation of this pathway in the absence of nitrate in the medium (see Discussion). We also observed downregulation of five light harvesting complex coding genes and PsbW (PF07123), suggesting a decrease in photosynthetic activity even though the alternative nitrogen source supported *P. calceolata* growth. Many upregulated genes under alternative nitrogen sources have unknown functions (13 out of 28 genes). We note the presence of three genes involved in protein folding (two Heat Shock Proteins and a Cyclophilin) and three genes involved in protein, fatty acid, and carbohydrate catabolism (a cysteine peptidase, a phytanoyl-CoA deoxygenase, and a glycosyl hydrolase family 16), suggesting cellular stress and activation of recycling in the absence of nitrate.

Only seven genes were specifically upregulated under cyanate (Fig. 4D; Table S5). Among these, one gene (Pca_1p27950) contains a domain of the transporter super-family FepB (COG0614; ABC-type $Fe3+$-hydroxamate transport system) and carries one transmembrane domain. An enzyme carrying an alkyl-hydroperoxide reductase domain (AhpD-like, IPR029032) was also upregulated under cyanate. This family of reductases is involved in defense against reactive oxygen species (56). The five other genes are a Triosephosphate isomerase (K01803-EC:5.3.1.1), a Glycosyl transferase of family 90 (PF05686), a proline iminopeptidase (K01259-EC:3.4.11.5), and two genes of unknown function.

## Nitrogenous compound transport

In each family of nitrogen compound transporters, we observed genes with important variations of expression when the nitrogen source or concentration was modified (Fig. 5 and 6; Table S5). The two high-affinity nitrate transporters (NRT2 and K02575) were downregulated when nitrate was replaced by another nitrogen source, but they were not regulated with the decrease of nitrate concentration. Two formate/nitrite transporters (FNT and PF01226) were downregulated in all alternative N sources. One FNT is a putative NAR1 transporter located on the chloroplast membrane. The other FNT was upregulated in low-nitrate conditions *in situ* and in laboratory cultures and putatively located on lysosome/vacuole membranes, suggesting that this gene may be involved in the transportation of recycled nitrite products from intracellular vacuoles. Among the five ammonium transporters (Amt), one gene was upregulated in low-nitrate conditions *in situ* and slightly upregulated in the intermediate-$NO_3$ condition, suggesting that this gene is a high-affinity transporter. This gene was also downregulated when nitrate was

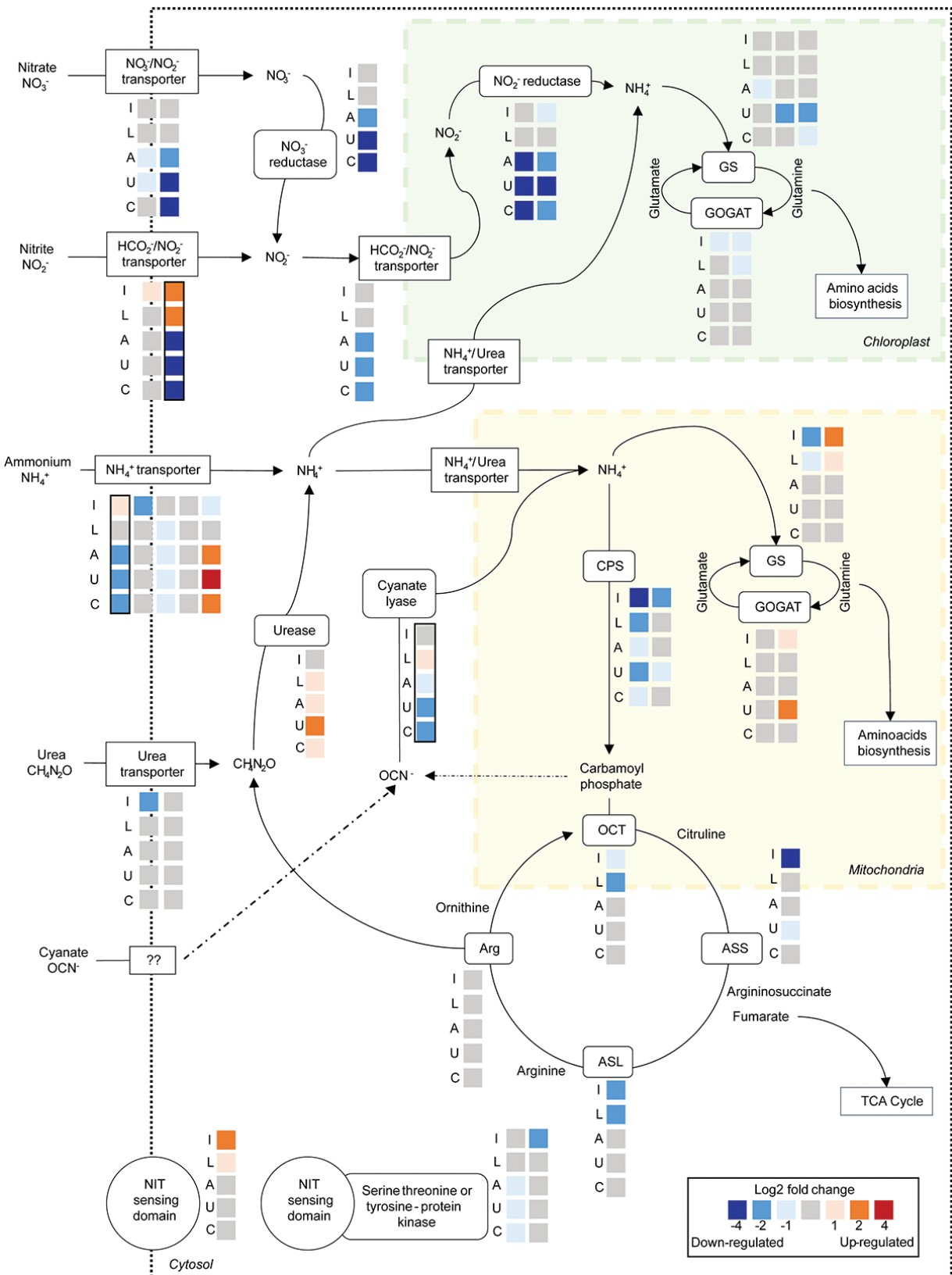

**FIG 6** Putative nitrogen metabolism in *P. calceolata* RCC100 under low-nitrate conditions or alternative nitrogen sources. Each square represents the differential expression of one gene in one condition compared to 882 µM nitrate. Each column is one gene, and rows are of different culture conditions. I: intermediate nitrate (441 µM); L: low-nitrate (220 µM); A: ammonium; U: urea; C: cyanate. GS: glutamine synthetase; GOGAT: glutamate synthase; CPS: carbamoyl-phosphate synthetase; OCT: ornithine-carbamoyl transferase; ASS: argininosuccinate synthase; ASL: argininosuccinate lyase; Arg: arginase. The color for each square indicates if the gene is upregulated (red) or downregulated (blue). Genes with black lines are upregulated in environmental low-nitrate samples.

replaced by another nitrogen source, including ammonium. Three other Amt genes were slightly downregulated when nitrate was depleted and might be low-affinity transporters. Interestingly, the last low-affinity transporter was strongly upregulated in urea, cyanate, and ammonium conditions, suggesting that this Amt is regulated according to the presence or absence of nitrate in the environment. One urea-proton symporter (K20989) exhibited an opposite pattern of expression in the two *P. calceolata* strains, with upregulation in low-nitrate in RCC697 and downregulation in RCC100. The other urea transporter (PF03253) in the *P. calceolata* genome was not differentially regulated in any of our tested conditions. Finally, the uric-acid/xanthine permease (K23887) upregulated in low-nitrate environments *in situ* was also significantly upregulated in low-nitrate cultures and downregulated when nitrate was replaced by cyanate or urea. This expression pattern suggests that this transporter recycles purine degradation products under nitrate starvation but does not transport extracellular nitrogenous molecules.

## Nitrate sensing

Three genes carrying a nitrate/nitrite sensing domain (NIT) are present in the *P. calceolata* genome. These genes had distinct expression patterns according to nitrate concentration or source (Fig. 5 and 6). The NIT-sensing gene carrying a transmembrane domain was upregulated in low-nitrate samples, but not differentially expressed when nitrate was replaced by another nitrogen source. This gene has been shown to be upregulated by *P. calceolata* in low-nitrate environments (32, 38), but was not significant in our environmental DESeq2 analysis. The two other nitrate-sensing genes carry a serine–threonine/tyrosine kinase domain and might play a role in phosphorylation-based signal transduction. One of these two genes was slightly downregulated in all alternative nitrogen sources, while the second exhibited an opposite pattern of expression in the two *P. calceolata* strains in reduced-nitrate conditions. This observation suggests that each gene has a specific role to respond to changes in intracellular or extracellular nitrate concentration.

## Nitrate reduction and storage

The NADH-dependent nitrate reductase (NR, K10534), the NADH-dependent nitrite reductase (NirB-NirD, PF01077) and the ferredoxin-dependent nitrite reductase (NirA, K00366) had the same expression pattern with higher gene expression levels when the nitrogen source was nitrate and strong downregulation under urea, cyanate, and ammonium (Fig. 5). Four of the five genes coding for GS in *P. calceolata* were differentially expressed in at least one experiment. Two GSs were downregulated when the nitrogen source was urea, suggesting that urea uptake and metabolism do not require these genes. Two putatively mitochondrial GSs were differentially expressed according to nitrate concentration in the laboratory experiments with opposite patterns. Four genes encode *P. calceolata* glutamate synthases (GOGAT). The putative Fd-GOGAT and an NADPH-GOGAT (GLT1, K00266), predicted to be located in the chloroplast according to peptide signal analysis (see Materials and Methods), were slightly downregulated in low-nitrate samples. Conversely, the putatively mitochondrial GOGAT was slightly upregulated in low-nitrate experiments (Fig. 5 and 6).

## Cyanate lyase, urease, and urea cycle

As in low-nitrate environmental samples, the *P. calceolata* cyanate lyase gene was slightly upregulated in our low-nitrate experiment in both strains (Table 1). Surprisingly, under alternative nitrogen sources, including cyanate, the cyanate lyase was downregulated (Fig. 5). This result shows that cyanate lyase is not involved in the assimilation of extracellular cyanate and could instead be involved in nitrogen recycling from intracellular molecules (see Discussion). *P. calceolata* encodes one urease (URE, K01427) and three urease accessory proteins: one UreD-UreF (K03190; K03188) and two UreG (PF02492). The urease was slightly upregulated in low-nitrate samples and upregulated in alternative

nitrogen sources, especially urea. This result indicates that extracellular urea as well as intracellularly produced urea are hydrolyzed into ammonia by the urease. The urea cycle in *P. calceolata* is composed of two carbamoyl phosphate synthetases (CPSs), one ornithine carbamoyltransferase (OCT), one arginosuccinate synthase (ASS), one arginosuccinate lyase (ASL), and one arginase (Fig. 5 and 6). Except for the arginase, all genes involved in the urea cycle were upregulated in high-nitrate conditions. Together with strong upregulation of the putatively mitochondrial GOGAT, this pattern indicates the removal of excess ammonia from the cell through the urea cycle.

## DISCUSSION

### *P. calceolata* has common patterns of low-nitrate acclimation *in situ* and in culture experiments

In this study, the regulation of the nitrogen metabolism under low-nitrate conditions and different nitrogen sources has been studied in the cosmopolitan and abundant microalgae *P. calceolata*. In addition, we used environmental metatranscriptomes from the *Tara* Oceans expedition to identify the genes that are regulated *in situ*. We identified only six genes significantly differentially expressed in low-nitrate environments *in situ*: three transporters of nitrogenous compounds (formate-nitrite, ammonium, and purine), a formyltetrahydrofolate deformylase, a cyanate lyase, and a gene of unknown function. These genes were also differentially expressed in low-nitrate conditions in the laboratory except for the ammonium transporter (Fig. 1C; Table 1). Extended periods of cultivation, several decades for *P. calceolata*, may expose strains to genomic modifications. However, the regulation of these genes in response to low nitrate has evidently been conserved compared to wild *Pelagomonas*.

Because of the relatively low abundance of cells in the environment, metatranscriptomes generally provide access only to highly expressed genes. Therefore, laboratory experiments are needed to observe fine-scale variations in genes with low expression levels, providing a more detailed picture of metabolic processes. Moreover, due to the multitude of environmental factors influencing gene expression levels in the environment, it is often challenging to disentangle the effects of individual parameters and accurately assess specific responses. In our analysis, we used the large diversity of environments sampled during the *Tara* cruise to get among low-nitrate samples a large range of all other parameters (temperature, salinity, iron concentrations, etc.). In this manner, the risk of environmental variables correlated with nitrate concentration is limited, and the differential expression analysis will only identify genes that were directly affected by nitrate concentrations. In environmental metatranscriptomes generated across large geographical areas, it can be challenging to determine whether gene expression variations are the result of acclimation (reversible short-term transcriptomic regulation) or adaptation (long-term selection) (38, 57). Complementing *in situ* analysis with culture-based transcriptomics allowed us to conclude that for genes that are differentially expressed in both situations, variations in *in situ* expression levels are likely the result of short-term acclimation.

### *P. calceolata* is genetically adapted to consume organic nitrogen compounds

We have shown that *P. calceolata* is capable of growth under inorganic (nitrate) or organic nitrogen source (urea or cyanate). The assimilation of nitrate by *P. calceolata* in oligotrophic areas was expected based on previous environmental and laboratory studies (38, 58, 59). Growth on urea as the sole nitrogen source aligns with existing literature, given the diversity of algae with this capacity, including other ochrophytes (10, 60, 61). The transcriptomic profile of *P. calceolata* underwent significant changes when cultured with organic nitrogen compounds, though stress response genes were not upregulated. In our results, we could disentangle a common transcriptomic response from a specific response to each nitrogen source. The main common response in the absence of nitrate was downregulation of genes of the nitrate assimilation pathway, such

as formate-nitrite and nitrate-nitrite transporters, as well as nitrate and nitrite reductases as shown in *P. tricornutum* (29). This pathway was not downregulated when nitrate concentration was reduced, suggesting that residual nitrate in our low-nitrate conditions was sufficient to maintain the pathway. The presence of urea negatively affects plastidic nitrogen assimilation in the pelagophyte strain CCMP2097 (62). In the pelagophyte *A. anophagefferens*, the use of urea as a nitrogen source triggers upregulation of genes involved in protein, amino acid, spermine, and sterol synthesis (28), but these functions were not upregulated in our experiments with *P. calceolata*.

## Recycling of intracellular nitrogenous compounds is the dominant response under low-nitrate conditions

As isolated algae can rapidly be derived genetically under laboratory conditions, we believe it is important to study different strains of the same taxa (63). In addition, different environmental *Pelagomonas* populations may have different acclimation capacities. Here, we worked with two strains of *P. calceolata* (RCC100 and RCC697) that are morphologically distinct (Fig. S1A and B), isolated from different oceans (Pacific and Indian Oceans) and at different times (1973 and 2003). We observed that RCC697 requires a stronger nitrate limitation (50 µm) compared to RCC100 (220 µm) to induce a transcriptomic response, but the gene expression patterns are similar. Our observed acclimation responses are therefore more likely to be general to all natural populations of *Pelagomonas*. The only difference between the two strains is the opposite transcriptomic regulation of a urea transporter (K20989) in low-nitrate conditions. This interesting pattern suggests that the strain isolated from the Pacific Ocean (RCC100) lost the ability to upregulate urea transporters, which is common in microalgae, but retained the capacity to grow under urea (64, 65).

Under low-nitrate conditions, both *P. calceolata* strains primarily rely on downregulating protein biosynthesis and recycling intracellular nitrogenous compounds such as amino acids and nucleotides, as previously suggested (59). Reducing nitrogen needs by decreasing carbon fixation, protein biosynthesis, and carbohydrates metabolism is quite common in diatoms, and gathering intracellular nitrogen in low-nitrate conditions was shown in *Aureococcus* (28, 65). A gene coding a xanthine/uracil/vitamin C permease (XUV), involved in the transport of purine degradation products, is upregulated both in environmental and experimental low-nitrate conditions. Upregulation of XUV under low-nitrate conditions has previously been observed in several microalgae, including *Aureococcus* and the haptophyte *Prymnesium parvum* (28, 66, 67). The expression of this permease in *P. calceolata* seems to also be linked to catabolism of purines as a recycling mechanism in low-N environments and does not reflect the presence of purines in the environment. Enzymes necessary for the conversion of xanthine into urea and ammonia are present in the *P. calceolata* genome, but were not upregulated in low-N conditions. This pathway is unlikely to be involved in purine recycling in *P. calceolata*, in contrast to *Aureococcus* when supplied with xanthine (68). Recent research highlights that many microalgae, including ochrophytes, are able to store nitrogen in purine crystals (69). Upregulation of a purine permease in *P. calceolata* under low-nitrate conditions could be a sign of nitrogen reallocation from crystals, but crystalline inclusions have not been reported in pelagophytes to date.

## Role of cyanate lyase in *P. calceolata*

Among eukaryotic microalgae, growth with cyanate as the sole nitrogen source has only previously been demonstrated in the dinoflagellate *Prorocentrum donghaiense*, albeit with a reduced growth rate (21). It has been suggested that phytoplankton upregulating cyanate lyase in low-nitrate environments are therefore capable of cyanate uptake and metabolism (26, 28). Although *P. calceolata* cyanate lyase is upregulated in low-nitrate environments (*in situ* and in culture), the bacterial community is required for *P. calceolata* to thrive under cyanate and that the cyanate lyase gene is downregulated in this case.

The available genomes of the two bacteria (*Paracoccus* sp. and *Marinobacter* sp.) identified in *P. calceolata* culture contain the CynS gene. These bacteria are theoretically capable of producing nitrogenous compounds from cyanate, which could be used by *P. calceolata*. Many compounds are involved in bacterial-algal cross-talks, including vitamins, signaling molecules, and nutrients (70–72). If the bacterial phycosphere is not required under standard culture conditions, different environments may require the processing of molecules by one or more bacteria to produce nutrients that support *P. calceolata* growth.

In consequence, cyanate lyase seems to not be involved in external cyanate metabolism and should not be used as a marker of cyanate uptake. In agreement with our results, cyanate supports the growth of several ascomycete species despite the absence of cyanate lyase coding genes in their genome (24). Conversely, several yeasts with cyanate lyase genes were unable to grow under cyanate, supporting the hypothesis that this gene is not involved in the assimilation of external cyanate in eukaryotes in contrast to prokaryotes. In *P. calceolata*, like all other eukaryotes, the bacterial cyanate transporter (CynX) is not conserved (73). Among the seven genes specifically upregulated in the sole presence of cyanate, a single gene carries homologies with a protein transporter domain (ABC-type $Fe^{3+}$-hydroxamate). Since this gene has only one transmembrane domain, it is probably not involved in cyanate uptake (Table S5).

The function of cyanate lyase in *P. calceolata* is likely the recycling of intracellularly produced cyanate. Cyanate can be generated through the rapid decomposition of CP (19, 74) or the slow decomposition of urea (75). *P. calceolata* recycles amino acids and proteins when nitrate supply is limited. Many enzymes are involved in the catabolism of metabolites in this process, including carbamoyltransferases (K00612) that can produce CP. Two of the five genes coding for carbamoyltransferases were upregulated in low-nitrate conditions in RCC100 (Fig. 5). We hypothesize that these enzymes increase the intracellular concentration of CP, which is in principle taken up by the urea cycle. However, under low-nitrate conditions, the urea cycle is downregulated, which could lead to an increase in intracellular cyanate, requiring cyanate lyase to produce ammonia. In complement to the work of Sato, Hashihama, and Takeda 2023 and the phylogeny of Mao et al. 2022, our results suggest that the removal of intracellular cyanate rather than external cyanate metabolism as an alternative N source is the main role of cyanate lyase in eukaryotic microalgae (26, 30).

## ACKNOWLEDGMENTS

The authors thank the commitment of the following people who made this work possible: the Genoscope/CEA, Paris-Saclay University and the CNRS; members of the Roscoff Culture Collection for maintaining *Pelagomonas* strains; Claude Scarpelli for support in high-performance computing at Genoscope. The authors acknowledge the financial support of the ANR (ANR-22-CE20-0012; ANR20-CE02-0025) and FRANCE GENOMIQUE (ANR-10-INBS-09-08). The authors also thank the *Tara* Expedition Foundation and their partners for the organization of marine scientific expeditions (http://oceans.taraexpeditions.org).

N.G., Q.C., and A.T. conceived and planned this study with the support of P.W., P.G., and I.P. N.G. and C.S. performed *P. calceolata* cultures with the strong support of C.O. and L.B. B.V., E.B., and G.M. did RNA extractions, library preparations, and sequencing coordinated by K.L. N.G. and C.S. carried out all bioinformatics analysis supervised by Q.C. N.G. and Q.C. wrote the paper. All authors contributed to the manuscript preparation and approved the final version of the paper.

## AUTHOR AFFILIATIONS

[1]Génomique Métabolique, Genoscope, Institut François Jacob, CEA, CNRS, Univ Evry, Université Paris-Saclay, Evry-Courcouronnes, France

[2]Research Federation for the Study of Global Ocean Systems Ecology and Evolution, R2022/Tara Oceans GO-SEE, Paris, France

[3]FR2424, Station Biologique de Roscoff, Sorbonne Université, CNRS, Roscoff, Brittany, France

[4]Genoscope, Institut François Jacob, CEA, Université Paris-Saclay, Evry-Courcouronnes, France

## PRESENT ADDRESS

Nina Guérin, Developmental Biology Unit, EMBL, Heidelberg, Germany

## AUTHOR ORCIDs

Quentin Carradec http://orcid.org/0000-0003-4612-8678

## FUNDING

| Funder | Grant(s) | Author(s) |
| --- | --- | --- |
| Agence Nationale de la Recherche | ANR-22-CE20-0012 | Quentin Carradec |
| Agence Nationale de la Recherche | ANR20-CE02-0025 | Ian Probert |
| Agence Nationale de la Recherche | ANR-10-INBS-09-08 | Patrick Wincker |

## AUTHOR CONTRIBUTIONS

Nina Guérin, Conceptualization, Data curation, Formal analysis, Investigation, Methodology, Visualization, Writing – original draft, Writing – review and editing | Chloé Seyman, Formal analysis, Investigation, Visualization, Writing – review and editing | Céline Orvain, Conceptualization, Data curation, Formal analysis, Investigation, Methodology, Resources, Visualization, Writing – original draft, Writing – review and editing | Laurie Bertrand, Methodology, Resources | Priscillia Gourvil, Methodology, Resources, Writing – review and editing | Ian Probert, Methodology, Writing – review and editing | Benoit Vacherie, Methodology, Resources | Élodie Brun, Methodology, Resources | Ghislaine Magdelenat, Methodology, Resources | Karine Labadie, Methodology, Resources, Supervision, Writing – review and editing | Patrick Wincker, Conceptualization, Funding acquisition, Supervision, Writing – review and editing | Adrien Thurotte, Conceptualization, Investigation, Methodology, Resources, Supervision, Validation, Writing – review and editing | Quentin Carradec, Conceptualization, Data curation, Formal analysis, Funding acquisition, Investigation, Methodology, Project administration, Resources, Supervision, Validation, Visualization, Writing – original draft, Writing – review and editing

## DATA AVAILABILITY

The transcriptomic data generated in this study have been deposited in the European Nucleotide Archive database under accession code PRJEB74085. The data that support the findings of this study are openly available in Zenodo at https://doi.org/10.5281/zenodo.12726053 and https://doi.org/10.5281/zenodo.6983364 for gene expression levels of *P. calceolata* in laboratory experiments and *Tara* Oceans metatranscriptomes. All codes for the bioinformatic workflow are provided at: https://github.com/institut-de-genomique/PelagomonasNitrogenMetabolism.

## ADDITIONAL FILES

The following material is available online.

## Supplemental Material

**Figures S1 and S2 (Spectrum02654-24-s0001.pdf).** Pelagomonas growth curves and gene expression levels between triplicates.

**Tables S1 to S6 (Spectrum02654-24-s0002.xlsx).** Data tables of sequencing statistics and gene expression analyses.

## Open Peer Review

**PEER REVIEW HISTORY (review-history.pdf).** An accounting of the reviewer comments and feedback.

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
