## [Reviewer comments · Microbiology Spectrum]

Microbiology Spectrum

Transcriptomic response of the picoalga *Pelagomonas calceolata* to nitrogen availability : New insights into cyanate lyase function.

Nina Guérin, Chloé Seyman, Céline Orvain, Laurie Bertrand, Priscillia Gourvil, Ian Probert, Karine Labadie, Elodie Brun, Ghislaine Magdelenat, Benoit Vacherie, Patrick Wincker, Adrien Thurotte, and Quentin Carradec

Corresponding Author(s): Quentin Carradec, Genomique Metabolique

Review Timeline:

Submission Date:	October 22, 2024
Editorial Decision:	December 9, 2024
Revision Received:	February 7, 2025
Accepted:	February 12, 2025

Editor: Adriana Lopes dos Santos

Reviewer(s): Disclosure of reviewer identity is with reference to reviewer comments included in decision letter(s). The following individuals involved in review of your submission have agreed to reveal their identity: Nastasia Johana Freyria (Reviewer #1)

Transaction Report:

DOI: <https://doi.org/10.1128/spectrum.02654-24>

Re: Spectrum02654-24 (Transcriptomic response of the picoalga *Pelagomonas calceolata* to nitrogen availability : New insights into cyanate lyase function.)

Dear Dr. Quentin Carradec:

Thank you for the privilege of reviewing your work. Below you will find my comments, instructions from the Spectrum editorial office, and the reviewer comments.

Revision Guidelines

Sincerely,
Adriana Lopes dos Santos
Editor
Microbiology Spectrum

Reviewer #1 (Comments for the Author):

The study on the picoalga *Pelagomonas* using culture-dependent transcriptomics under varying concentrations and types of nitrogen is intriguing and provides new insights into the function of cyanate lyase. The authors present a strong approach by exposing two strains of *Pelagomonas* to different nitrogen sources and incorporating a large dataset of metatranscriptomic samples from the Tara Oceans project. However, the comparison between the two strains is limited, and the rationale for choosing this specific species is not sufficiently emphasized. Additionally, more rigorous statistical tests are necessary to

validate the reported significance or lack thereof in certain results (as mentioned below).

Furthermore, identifying the microbial communities in non-axenic cultures using basic methods such as 16S rRNA gene amplicon sequencing could be valuable. This approach might also facilitate the annotation of functional genes within the microbial community, providing deeper insights into the interactions and functional potential of the associated microorganisms. Overall, the study is interesting and shows good results. However, the authors should address the following points for clarification:

Abstract:

- L. 27: Typos error, replace "in" by "is".
- L. 26: Typos error, add an "s" at "describe".

Introduction:

- L. 55: Typos error, add "of" after "average".
- L. 71-72: Please cite a literature reference for urea transporter present in these two mentioned species.
- L. 92: Start the sentence with "The 15N".
- L. 110: Authors write that they cultivated 2 strains, but only one strain name was mentioned at L. 105. Please write the second strain name.

The main comment on the introduction is that the authors did not sufficiently highlight the ecological importance of pelagophytes in marine environments, particularly their role and abundance. While the authors mention several phytoplankton species capable of assimilating cyanate as well as dissolved inorganic nitrogen (DIN) and dissolved organic nitrogen (DON), they do not place enough emphasis on the specific target strain they studied. Key contextual details are missing: what type of marine environment this species inhabits, whether it is common, and if it is associated with temperate or polar waters. Additionally, the rationale for focusing on this particular species remains unclear.

Methods:

- L. 147: The authors should begin the sentence with a phrase such as "A volume of 1 mL" rather than starting directly with "1 mL."
- L. 158-159: The authors referenced Figure 3b before referencing all previous figures. Please remove the citation of Figure 3b from the Methods section.
- L. 159: Please confirm whether strain RCC697 was treated with antibiotics. If this was not the case, could you please explain the reason why? This information is not clear to the reader. Please provide more precise details about both strains' cultures for axenicity and non-axenicity, as well as the rationale behind this decision.
- L. 198: Please mention which version of SILVA database the authors used.
- L. 203: Please do not begin the sentence with a number; instead, add a linking word.
- L. 206: Please cite which package the cor function is used from.

Results:

- L. 241: Please clarify why the authors aligned all metatranscriptomes to only RCC100 and not RCC697. This aspect is not addressed in the manuscript. The authors should provide a clear rationale for this decision.
- Figure 1 legend: Acronyms from the figure should be mentioned in the title legend, e.g. Sur and DCM should be specified in the title legend.
- L. 247: Typos error, delete the space between "Figure 1B" and ";".
- L. 248: The authors should write "analyses" and not "analysis".
- L. 252-253: There are two citation of Guerin et al 2022. Please see comment on reference style below. The authors did not follow the Microbiology Spectrum reference style!
- L. 248-263: The entire paragraph discusses specific genes; however, it is difficult to locate these genes in Supplementary Table 2 or Figure 1B. None of the mentioned genes are highlighted in the main figures, making it challenging to observe the differential expression of these target genes compared to others. The purpose of Figure 1B is unclear, as the authors highlight only a few genes between DCM and Surface without emphasizing the specific genes mentioned in the paragraph at lines 248-263. Additionally, the authors provide minimal description of Figure 1B.
- Figure 1C is not mentioned at all!
- L. 280-282: Please mention Supplementary Figure 1C and D here!
- L. 286: Please do not begin the sentence with a number in percentage; instead, add a linking word.
- L. 287: add "of" after "average".
- L. 288-292: Please mention Supplementary Figure 2 in the first sentence at L. 288-290.
- L. 293-297: The number of DEGs for each comparison should also be indicated in Figure 2A, B, C, and D to help the reader more easily follow along with both the figure and the text.
- L. 300: Did the authors performed a statistical test that shows no significance? If yes, please add details, if no remove the significance word.
- L. 302: The authors should highlight the 95 common DEGs between the two strains in Supplementary Table 4. With 2,480 rows in the table, it would be challenging for readers to locate the information emphasized in the text.
- Figure 2F is not mentioned at all and therefore not described, before authors mentions Figure 3!
- L. 326-329: The all sentence after "indicating" should be moved to Discussion section.
- L. 341-342: Please specify the figure or table where readers can find the observation mentioned by the authors.

- L. 341-346: It would be helpful to add the number of DEGs for each comparison directly in Figure 4A, B, and C to assist the reader in following the results more easily.
- L. 348: The authors mentioned only twice LHC. I suggest authors to write LHC entirely twice since it is not mention several times in the manuscript.
- L. 348-350: Where do the authors observe the 6 LHC genes? There is no figure or table referenced in the text, and the LHC genes are not highlighted in Figure 4.
- L. 351-354: Same question as above, where can we see those mentioned genes?
- L. 362: Typos error, delete space between "COG0614" and ";".
- L. 371, 400, 414, 424: Typos error, please add an "s" to "Figure 5, 6".
- L. 411-414: Genes NR, NirB, NirD and NirA are present only in Figure 5 and not Figure 6. Please mention only Figure 5 in the sentence. Or explain where these mentioned genes are in Figure 6.
- L. 414-417: This sentence should be moved to Discussion section.
- L. 422-423: On what basis do the authors state that the mentioned genes are predicted to be located in the chloroplast? Did the authors perform signal peptide analyses, or is this prediction based on literature? If the latter, please provide appropriate citations.
- L. 436-437: Please mention which figure the results is observable.
- L. 440: Typo error, add a space between the two KO numbers.
- L. 443-445: Where in the text do the authors indicate that the urea cycle consists of all the genes shown in Figure 4, as referenced? Could the authors have cited the wrong figure?

Discussion:

- L.452: When the authors claim that this is the first study on nitrogen sources in pelagophytes, they should be aware that transcriptomic analyses of another pelagophyte, especially CCMP2097, along with several other microalgae, were conducted as part of the MNETSP project in 2016 (Keeling et al., 2016, 10.1371/journal.pbio.1001889). The authors should either revise their statement or cite this work.
- L. 454-456: Please cite the names of the 6 genes and add the figure or table where results come from.
- L. 471-472: The authors' statement is valid, but they should also cite previous transcriptomic studies on known pelagophytes or related microalgae, particularly those focused on acclimatization and adaptation.
- L.486: The authors should provide the full genus name of *Auerococcus*, as it has not been mentioned earlier in the paragraph. This will help readers who are unfamiliar with the genus to better follow the text.
- L. 522-523: Please cite the figures where results come from in the Discussion.

The discussion section is too brief and leaves readers wanting more. There is no meaningful comparison between the two strains, nor an explanation of why the authors chose to culture two different strains from the same genus. Additionally, the discussion lacks a comparison with previous studies. The reference list is relatively short, with only about 50 references in total, and fewer than 15 are cited in the discussion section. It would have been valuable to integrate information on the types of bacteria coexisting with the strains in culture. Why did the authors not identify the bacteria present in the non-axenic cultures, one basic technique is the use of 16S rRNA gene sequencing?

References list:

It is the responsibility of the authors to recheck every reference to ensure that all Latin names are in italics (e.g. errors were found for Guérin et al.), and that the title should not be in italics (e.g. errors were found for second reference of Guérin et al.). The authors did not adhere to the Microbiology Spectrum reference style, which requires each reference to be numbered.

The study on the picoalga *Pelagomonas* using culture-dependent transcriptomics under varying concentrations and types of nitrogen is intriguing and provides new insights into the function of cyanate lyase. The authors present a strong approach by exposing two strains of *Pelagomonas* to different nitrogen sources and incorporating a large dataset of metatranscriptomic samples from the Tara Oceans project. However, the comparison between the two strains is limited, and the rationale for choosing this specific species is not sufficiently emphasized. Additionally, more rigorous statistical tests are necessary to validate the reported significance or lack thereof in certain results (as mentioned below).

Furthermore, identifying the microbial communities in non-axenic cultures using basic methods such as 16S rRNA gene amplicon sequencing could be valuable. This approach might also facilitate the annotation of functional genes within the microbial community, providing deeper insights into the interactions and functional potential of the associated microorganisms.

Overall, the study is interesting and shows good results. However, the authors should address the following points for clarification:

Abstract:

- **L. 27.** Typos error, replace “in” by “is”.
- **L. 26:** Typos error, add an “s” at “describe”.

Introduction:

- **L. 55:** Typos error, add “of” after “average”.
- **L. 71-72:** Please cite a literature reference for urea transporter present in these two mentioned species.
- **L. 92:** Start the sentence with “The 15N”.
- **L. 110:** Authors write that they cultivated 2 strains, but only one strain name was mentioned at **L. 105**. Please write the second strain name.

The main comment on the introduction is that the authors did not sufficiently highlight the ecological importance of pelagophytes in marine environments, particularly their role and abundance. While the authors mention several phytoplankton species capable of assimilating cyanate as well as dissolved inorganic nitrogen (DIN) and dissolved organic nitrogen (DON), they do not place enough emphasis on the specific target strain they studied. Key contextual details are missing: what type of marine environment this species inhabits, whether it is common, and if it is associated with temperate or polar waters. Additionally, the rationale for focusing on this particular species remains unclear.

Methods:

- **L. 147:** The authors should begin the sentence with a phrase such as “A volume of 1 mL” rather than starting directly with “1 mL.”

- **L. 158-159:** The authors referenced Figure 3b before referencing all previous figures. Please remove the citation of Figure 3b from the Methods section.
- **L. 159:** Please confirm whether strain RCC697 was treated with antibiotics. If this was not the case, could you please explain the reason why? This information is not clear to the reader. Please provide more precise details about both strains' cultures for axenicity and non-axenicity, as well as the rationale behind this decision.
- **L. 198:** Please mention which version of SILVA database the authors used.
- **L. 203:** Please do not begin the sentence with a number; instead, add a linking word.
- **L. 206:** Please cite which package the cor function is used from.

Results:

- **L. 241:** Please clarify why the authors aligned all metatranscriptomes to only RCC100 and not RCC697. This aspect is not addressed in the manuscript. The authors should provide a clear rationale for this decision.
- **Figure 1 legend:** Acronyms from the figure should be mentioned in the title legend, e.g. Sur and DCM should be specified in the title legend.
- **L. 247:** Typos error, delete the space between "Figure 1B" and ",".
- **L. 248:** The authors should write "analyses" and not "analysis".
- **L. 252-253:** There are two citation of Guerin et al 2022. Please see comment on reference style below. The authors did not follow the Microbiology Spectrum reference style!
- **L. 248-263:** The entire paragraph discusses specific genes; however, it is difficult to locate these genes in Supplementary Table 2 or Figure 1B. None of the mentioned genes are highlighted in the main figures, making it challenging to observe the differential expression of these target genes compared to others. The purpose of Figure 1B is unclear, as the authors highlight only a few genes between DCM and Surface without emphasizing the specific genes mentioned in the paragraph at lines 248–263. Additionally, the authors provide minimal description of Figure 1B.
- **Figure 1C** is not mentioned at all!
- **L. 280-282:** Please mention Supplementary Figure 1C and D here!
- **L. 286:** Please do not begin the sentence with a number in percentage; instead, add a linking word.
- **L. 287:** add "of" after "average".
- **L. 288-292:** Please mention Supplementary Figure 2 in the first sentence at **L. 288-290**.
- **L. 293-297:** The number of DEGs for each comparison should also be indicated in Figure 2A, B, C, and D to help the reader more easily follow along with both the figure and the text.
- **L. 300:** Did the authors performed a statistical test that shows no significance? If yes, please add details, if no remove the significance word.
- **L. 302:** The authors should highlight the 95 common DEGs between the two strains in Supplementary Table 4. With 2,480 rows in the table, it would be challenging for readers to locate the information emphasized in the text.

- **Figure 2F** is not mentioned at all and therefore not described, before authors mentions Figure 3!
- **L. 326-329:** The all sentence after “indicating” should be moved to Discussion section.
- **L. 341-342:** Please specify the figure or table where readers can find the observation mentioned by the authors.
- **L. 341-346:** It would be helpful to add the number of DEGs for each comparison directly in Figure 4A, B, and C to assist the reader in following the results more easily.
- **L. 348:** The authors mentioned only twice LHC. I suggest authors to write LHC entirely twice since it is not mention several times in the manuscript.
- **L. 348-350:** Where do the authors observe the 6 LHC genes? There is no figure or table referenced in the text, and the LHC genes are not highlighted in Figure 4.
- **L. 351-354:** Same question as above, where can we see those mentioned genes?
- **L. 362:** Typos error, delete space between “COG0614” and “;”.
- **L. 371, 400, 414, 424:** Typos error, please add an “s” to “Figure 5, 6”.
- **L. 411-414:** Genes NR, NirB, NirD and NirA are present only in Figure 5 and not Figure 6. Please mention only Figure 5 in the sentence. Or explain where these mentioned genes are in Figure 6.
- **L. 414-417:** This sentence should be moved to Discussion section.
- **L. 422-423:** On what basis do the authors state that the mentioned genes are predicted to be located in the chloroplast? Did the authors perform signal peptide analyses, or is this prediction based on literature? If the latter, please provide appropriate citations.
- **L. 436-437:** Please mention which figure the results is observable.
- **L. 440:** Typo error, add a space between the two KO numbers.
- **L. 443-445:** Where in the text do the authors indicate that the urea cycle consists of all the genes shown in Figure 4, as referenced? Could the authors have cited the wrong figure?

Discussion:

- **L.452:** When the authors claim that this is the first study on nitrogen sources in pelagophytes, they should be aware that transcriptomic analyses of another pelagophyte, especially CCMP2097, along with several other microalgae, were conducted as part of the MMETSP project in 2016 (Keeling et al., 2016, 10.1371/journal.pbio.1001889). The authors should either revise their statement or cite this work.
- **L. 454-456:** Please cite the names of the 6 genes and add the figure or table where results come from.
- **L. 471-472:** The authors' statement is valid, but they should also cite previous transcriptomic studies on known pelagophytes or related microalgae, particularly those focused on acclimatization and adaptation.
- **L.486:** The authors should provide the full genus name of *Auerococcus*, as it has not been mentioned earlier in the paragraph. This will help readers who are unfamiliar with the genus to better follow the text.

- **L. 522-523:** Please cite the figures where results come from in the Discussion.

The discussion section is too brief and leaves readers wanting more. There is no meaningful comparison between the two strains, nor an explanation of why the authors chose to culture two different strains from the same genus. Additionally, the discussion lacks a comparison with previous studies. The reference list is relatively short, with only about 50 references in total, and fewer than 15 are cited in the discussion section. It would have been valuable to integrate information on the types of bacteria coexisting with the strains in culture. Why did the authors not identify the bacteria present in the non-axenic cultures, one basic technique is the use of 16S rRNA gene sequencing?

References list:

It is the responsibility of the authors to recheck every reference to ensure that all Latin names are in italics (e.g. errors were found for Guérin et al.), and that the title should not be in italics (e.g. errors were found for second reference of Guérin et al.). The authors did not adhere to the Microbiology Spectrum reference style, which requires each reference to be numbered.

Reviewer #1 (Comments for the Author):

The study on the picoalga *Pelagomonas* using culture-dependent transcriptomics under varying concentrations and types of nitrogen is intriguing and provides new insights into the function of cyanate lyase. The authors present a strong approach by exposing two strains of *Pelagomonas* to different nitrogen sources and incorporating a large dataset of metatranscriptomic samples from the Tara Oceans project. However, the comparison between the two strains is limited, and the rationale for choosing this specific species is not sufficiently emphasized. Additionally, more rigorous statistical tests are necessary to validate the reported significance or lack thereof in certain results (as mentioned below).

Furthermore, identifying the microbial communities in non-axenic cultures using basic methods such as 16S rRNA gene amplicon sequencing could be valuable. This approach might also facilitate the annotation of functional genes within the microbial community, providing deeper insights into the interactions and functional potential of the associated microorganisms.

Overall, the study is interesting and shows good results. However, the authors should address the following points for clarification:

We thank the reviewer for the corrections and interesting comments and suggestions. We have improved the manuscript with several additions regarding the differences between the 2 strains, the bacteria identification and have enriched the discussion section. The details are below. Line numbers refer to the "Marked-Up Manuscript" file.

Abstract:

-L. 27. Typos error, replace "in" by "is".

Typo corrected.

-L. 26: Typos error, add an "s" at "describe".

Typo corrected.

Introduction:

-L. 55: Typos error, add "of" after "average".

Typo corrected.

-L. 71-72: Please cite a literature reference for urea transporter present in these two mentioned species.

We added two references for *Phaeodactylum* (Alipanah et al. 2015 ; <https://doi.org/10.1093/jxb/erv340>) et *Thalassiosira* (Chen et al 2018 ; <https://doi.org/10.3389/fmicb.2018.02761>) line 72.

-L. 92: Start the sentence with "The 15N".

Typo corrected.

-L. 110: Authors write that they cultivated 2 strains, but only one strain name was mentioned at L. 105. Please write the second strain name.

Strain name added line 112.

The main comment on the introduction is that the authors did not sufficiently highlight the ecological importance of pelagophytes in marine environments, particularly their role and

abundance. While the authors mention several phytoplankton species capable of assimilating cyanate as well as dissolved inorganic nitrogen (DIN) and dissolved organic nitrogen (DON), they do not place enough emphasis on the specific target strain they studied. Key contextual details are missing: what type of marine environment this species inhabits, whether it is common, and if it is associated with temperate or polar waters. Additionally, the rationale for focusing on this particular species remains unclear.

We added several sentences to introduce the diversity and the ecological importance of pelagophytes in the oceans line 107 to 114 :

“Pelagophyceae are a diverse group of marine microalgae comprising 4 families and 23 genera (33). They cover all oceanic basins, from polar waters to tropical oceans (34). Most species have been described in coastal environments some of them forming brown tides (35). The ability to consume organic nitrogen has been shown to contribute to *Aureococcus* blooms (36). Among the few pelagophytes present in the open ocean, *Pelagomonas* is the dominant taxa and widely distributed in temperate and tropical oceans (32). Environmental studies have shown that *Pelagomonas* present strong acclimation abilities, especially to iron and nitrate depletion (37, 38).”

Methods:

-L. 147: The authors should begin the sentence with a phrase such as "A volume of 1 mL" rather than starting directly with "1 mL."

Text added line 149.

-L. 158-159: The authors referenced Figure 3b before referencing all previous figures. Please remove the citation of Figure 3b from the Methods section.

Reference to the figure removed.

-L. 159: Please confirm whether strain RCC697 was treated with antibiotics. If this was not the case, could you please explain the reason why? This information is not clear to the reader. Please provide more precise details about both strains' cultures for axenic and non-axenic, as well as the rationale behind this decision.

Both *Pelagomonas* strains are usually maintained in non-axenic conditions. We choose to keep these conditions to be closer to the physiological condition of *Pelagomonas* growth in the environment and avoid growth trouble if the bacterial community provides important molecules to the algae.

To understand if *Pelagomonas* itself or the bacterial community is able to assimilate cyanate, we treated one strain (RCC100) with antibiotics (for the experiment presented in figure 3 only). This experiment of the different nitrogen sources was not performed on RCC697. To clarify the method, we modified the sentences line 144 and 159.

-L. 198: Please mention which version of SILVA database the authors used.

Version of SILVA database added.

-L. 203: Please do not begin the sentence with a number; instead, add a linking word.

Sentence modified

-L. 206: Please cite which package the cor function is used from.

Version added line 213

Results:

-L. 241: Please clarify why the authors aligned all metatranscriptomes to only RCC100 and not RCC697. This aspect is not addressed in the manuscript. The authors should provide a clear rationale for this decision.

We used RCC100 genome because RCC697 genome has not yet been sequenced. We clarified this point in the Mat&Met line 216.

-Figure 1 legend: Acronyms from the figure should be mentioned in the title legend, e.g. Sur and DCM should be specified in the title legend.

We modified the figure legend accordingly.

-L. 247: Typos error, delete the space between "Figure 1B" and ";".

Typo corrected

-L. 248: The authors should write "analyses" and not "analysis".

Typo corrected

-L. 252-253: There are two citation of Guerin et al 2022. Please see comment on reference style below. The authors did not follow the Microbiology Spectrum reference style!

The 2 citations of Guérin et al 2022 are different. The first is a dataset published in Zenodo and the second is an article. We removed the references to public datasets to conform with Microbiology Spectrum style.

-L. 248-263: The entire paragraph discusses specific genes; however, it is difficult to locate these genes in Supplementary Table 2 or Figure 1B. None of the mentioned genes are highlighted in the main figures, making it challenging to observe the differential expression of these target genes compared to others. The purpose of Figure 1B is unclear, as the authors highlight only a few genes between DCM and Surface without emphasizing the specific genes mentioned in the paragraph at lines 248-263. Additionally, the authors provide minimal description of Figure 1B.

The genes described in the paragraph are highlighted in Figure 1C. The mention to Figure 1B is a mistake in this paragraph and has been replaced by Figure 1C. Supplementary Table 2 contains all the differentially expressed genes with detailed functional information (395 lines).

-Figure 1C is not mentioned at all!

We corrected this mistake (see above)

-L. 280-282: Please mention Supplementary Figure 1C and D here!

We added the reference to the Supplementary Figure.

-L. 286: Please do not begin the sentence with a number in percentage; instead, add a linking word.

We modified the sentence line 290: "The two strains are genetically close with an average of 96.0% of nucleotide identity for 86% of RCC100 genes covered by at least one RCC697 read (Table S3)."

-L. 287: add "of" after "average".

Sentence modified (see above).

-L. 288-292: Please mention Supplementary Figure 2 in the first sentence at L. 288-290.
Mention moved at the end of the first sentence.

-L. 293-297: The number of DEGs for each comparison should also be indicated in Figure 2A, B, C, and D to help the reader more easily follow along with both the figure and the text.
Number of DEGs added in each panel.

-L. 300: Did the authors performed a statistical test that shows no significance? If yes, please add details, if no remove the significance word.
We replaced “significantly” with “strongly” line 305: “For RCC697, only 4 genes were differentially expressed in the 220 μ M condition, indicating that this nitrate reduction did not strongly affects this strain in contrast to RCC100”

- L. 302: The authors should highlight the 95 common DEGs between the two strains in Supplementary Table 4. With 2,480 rows in the table, it would be challenging for readers to locate the information emphasized in the text.
We added a column in Supplementary Table 4 named “DEG in the 2 strains” with “Yes” or “No” values. The 95 genes can be easily located with this new column. We corrected this number to 97.

- Figure 2F is not mentioned at all and therefore not described, before authors mentions Figure 3!
We added the mention of Figure 2F line 307.

- L. 326-329: The all sentence after "indicating" should be moved to Discussion section.
We changed the first sentence to “Growth under cyanate without the bacterial in axenic conditions was strongly reduced, indicating that RCC100 cannot metabolise cyanate without the bacterial community.” and removed the second sentence that is already present in the Discussion section.

- L. 341-342: Please specify the figure or table where readers can find the observation mentioned by the authors.
We added the reference of Figure 4A.

- L. 341-346: It would be helpful to add the number of DEGs for each comparison directly in Figure 4A, B, and C to assist the reader in following the results more easily.
We modified the figure similarly to Figure 2.

- L. 348: The authors mentioned only twice LHC. I suggest authors to write LHC entirely twice since it is not mention several times in the manuscript.
We removed the LHC abbreviation line 357.

- L. 348-350: Where do the authors observe the 6 LHC genes? There is no figure or table referenced in the text, and the LHC genes are not highlighted in Figure 4.
We added a Supplementary table (Table S6) to highlight these 65 genes differentially expressed in the 3 nitrogen conditions. The genes involved in photosynthesis are in bold (5 LHC and Psbw, corrected in the text line 357).

- L. 351-354: Same question as above, where can we see those mentioned genes?

New Supplementary Table 6 (see above).

- L. 362: Typos error, delete space between "COG0614" and ";".

Typo corrected

- L. 371, 400, 414, 424: Typos error, please add an "s" to "Figure 5, 6".

"s" added in the 4 sentences.

- L. 411-414: Genes NR, NirB, NirD and NirA are present only in Figure 5 and not Figure 6. Please mention only Figure 5 in the sentence. Or explain where these mentioned genes are in Figure 6.

We removed the mention of Figure 6

- L. 414-417: This sentence should be moved to Discussion section.

Sentence removed from the result and added lines 496-497.

- L. 422-423: On what basis do the authors state that the mentioned genes are predicted to be located in the chloroplast? Did the authors perform signal peptide analyses, or is this prediction based on literature? If the latter, please provide appropriate citations.

This is based on signal peptide analysis. The methodology is explained lines 241-243. We added "according to peptide signal analysis (see Method)" line 416-418.

- L. 436-437: Please mention which figure the results is observable.

We added the mention to Figure 5.

- L. 440: Typo error, add a space between the two KO numbers.

Typo corrected.

- L. 443-445: Where in the text do the authors indicate that the urea cycle consists of all the genes shown in Figure 4, as referenced? Could the authors have cited the wrong figure?

It's indeed a mistake. We changed the reference from Figure 4 to Figures 5 and 6.

Discussion:

- L.452: When the authors claim that this is the first study on nitrogen sources in pelagophytes, they should be aware that transcriptomic analyses of another pelagophyte, especially CCMP2097, along with several other microalgae, were conducted as part of the MMETSP project in 2016 (Keeling et al., 2016, 10.1371/journal.pbio.1001889). The authors should either revise their statement or cite this work.

This sentence states that this study is the first focusing on response to different nitrogen sources in the species *Pelagomonas calceolata*, not in the Pelagophyceae class. Strain CCMP2097 belongs to Plocamiomonas genus, recently described in this study : <https://doi.org/10.1080/09670262.2024.2353940> .

To our knowledge, the only study on *P. calceolata* from MMETSP transcriptome is the work of Kang et al 2021 (<https://doi.org/10.3389/fmars.2021.636699>). They did a low-N experiment but without replication and they did not test different nitrogen sources. Yet, we removed "for the first time" and added a citation of this work that also suggests the use of organic N in *P. calceolata* based on gene expression variations (line 441).

- L. 454-456: Please cite the names of the 6 genes and add the figure or table where results come from.

We added the names and rephrased the sentence lines 443-448: “We identified only 6 genes significantly differentially expressed in low-nitrate environments in situ : 3 transporters of nitrogenous compounds (formate-nitrite, ammonium and purine), a formyltetrahydrofolate deformylase, a cyanate lyase and a gene of unknown function. These genes were also differentially expressed in low-nitrate conditions in the laboratory except the ammonium transporter (Fig.1C and Table 1). “

- L. 471-472: The authors' statement is valid, but they should also cite previous transcriptomic studies on known pelagophytes or related microalgae, particularly those focused on acclimatization and adaptation.

We added the citation of an article that observed in situ gene expression variation of *Pelagomonas* but did not disentangle between acclimation and adaptation (Dupont et al 2015, <https://doi.org/10.1038/ismej.2014.198>) and a recent study on the acclimation of an arctic Pelagophyte (Freyria et al 2024, <https://doi.org/10.1038/s42003-024-06765-7>)

- L.486: The authors should provide the full genus name of *Auerococcus*, as it has not been mentioned earlier in the paragraph. This will help readers who are unfamiliar with the genus to better follow the text.

We added the genus name and removed the species name when not needed.

- L. 522-523: Please cite the figures where results come from in the Discussion.

This result comes from Figure 4D and the gene names and functional annotation are in Sup Table 5. We added a column “DEG only in the tested condition” to easily find the genes specific to each condition.

The discussion section is too brief and leaves readers wanting more. There is no meaningful comparison between the two strains, nor an explanation of why the authors chose to culture two different strains from the same genus. Additionally, the discussion lacks a comparison with previous studies. The reference list is relatively short, with only about 50 references in total, and fewer than 15 are cited in the discussion section.

We added in the discussion a paragraph to justify the choice of 2 *Pelagomonas* strains and the main difference observed between them lines 486 to 497:

“As isolated algae can rapidly derive genetically under laboratory conditions, we believe it is important to study different strains of the same taxa (63). In addition, different environmental *Pelagomonas* populations may have different acclimation capacities. Here, we worked with 2 strains of *P. calceolata* (RCC100 and RCC697) that are morphologically distinct (Fig. S1A and B), isolated from different oceans (Pacific and Indian Oceans) and at different times (1973 and 2003). We observed that RCC697 requires a stronger nitrate limitation (50µm) compared to RCC100 (220µm) to induce a transcriptomic response but the gene expression patterns are similar. Our observed acclimation responses are therefore more likely to be general to all natural populations of *Pelagomonas*. The only difference between the 2 strains is the opposite transcriptomic regulation of a urea transporter (K20989) in low-nitrate condition. This interesting pattern suggests that the strain isolated from the Pacific Ocean

(RCC100) lost the ability to upregulate urea transporters, which is common in microalgae, but retained the capacity to grow under urea (64, 65). “

We modified the discussion to add more comparisons with previous studies :

lines 468-471, we added :

“The assimilation of nitrate by *P. calceolata* in oligotrophic areas was expected based on previous environmental and laboratory studies (38, 58, 59). Growth on urea as the sole nitrogen source aligns with existing literature, given the diversity of algae with this capacity, including other ochrophytes (10, 60, 61).”

Lines 500-502 we added:

(59). Reducing nitrogen needs by decreasing carbon fixation, protein biosynthesis and carbohydrates metabolism is quite common in diatoms and gathering intracellular nitrogen in low-nitrate condition was shown in *Aureococcus* (28, 65).

It would have been valuable to integrate information on the types of bacteria coexisting with the strains in culture. Why did the authors not identify the bacteria present in the non-axenic cultures, one basic technique is the use of 16S rRNA gene sequencing?

To answer this question we isolated on plate, several bacteria present in the *P. calceolata* culture. We identified 2 bacteria, *Paracoccus* (alphaproteobacteria) and *Marinobacter* (gammaproteobacteria). We searched for cyanate lyase genes into the genomes available for this 2 genera and found that all genomes available for these 2 genera have the cyanate lyase (CynS) and cyanate transporter CynX genes. It is therefore likely that these bacteria convert Cyanate into a nitrogenous molecule that is then metabolised by *P. calceolata*. We added this new experiment and results on the manuscript lines 177 to 189 for the Methodology and line 327-334 in the result section. The crosstalk between *Pelagomonas* and its phycosphere is discussed lines 524 to 530.

References list:

It is the responsibility of the authors to recheck every reference to ensure that all Latin names are in italics (e.g. errors were found for Guérin et al.), and that the title should not be in italics (e.g. errors were found for second reference of Guérin et al.). The authors did not adhere to the Microbiology Spectrum reference style, which requires each reference to be numbered.

We changed the bibliography format according to Microbiology Spectrum style.

Re: Spectrum02654-24R1 (Transcriptomic response of the picoalga *Pelagomonas calceolata* to nitrogen availability : New insights into cyanate lyase function.)

Dear Dr. Quentin Carradec:

As you noticed, since an additional reviewer was not found as the editor, I have acted as the second review to expedite the process. I believe you accomplished all the important modifications and the data provided in your manuscript will be relevant to those in the field. Therefore, your manuscript has been accepted, and I am forwarding it to the ASM production staff for publication.

Your paper will first be checked to make sure all elements meet the technical requirements. ASM staff will contact you if anything needs to be revised before copyediting and production can begin. Otherwise, you will be notified when your proofs are ready to be viewed.

Sincerely,
Adriana Lopes dos Santos
Editor
Microbiology Spectrum